# Substrate-induced conformational dynamics of the dopamine transporter

Anne Kathrine Nielsen [1,2], Ingvar R. Möller [1,2], Yong Wang [3], Søren G.F. Rasmussen[4], Kresten Lindorff-Larsen [3], Kasper D. Rand [2] & Claus J. Loland [1]

The dopamine transporter is a member of the neurotransmitter:sodium symporters (NSSs), which are responsible for termination of neurotransmission through $Na^+$-driven reuptake of neurotransmitter from the extracellular space. Experimental evidence elucidating the coordinated conformational rearrangements related to the transport mechanism has so far been limited. Here we probe the global $Na^+$- and dopamine-induced conformational dynamics of the wild-type *Drosophila melanogaster* dopamine transporter using hydrogen-deuterium exchange mass spectrometry. We identify $Na^+$- and dopamine-induced changes in specific regions of the transporter, suggesting their involvement in protein conformational transitions. Furthermore, we detect ligand-dependent slow cooperative fluctuations of helical stretches in several domains of the transporter, which could be a molecular mechanism that assists in the transporter function. Our results provide a framework for understanding the molecular mechanism underlying the function of NSSs by revealing detailed insight into the state-dependent conformational changes associated with the alternating access model of the dopamine transporter.

[1] Laboratory for Membrane Protein Dynamics, Department of Neuroscience, University of Copenhagen, 2200 Copenhagen N, Denmark. [2] Protein Analysis Group, Department of Pharmacy, University of Copenhagen, 2100 Copenhagen Ø, Denmark. [3] Structural Biology and NMR Laboratory, Linderstrøm-Lang Centre for Protein Science, Department of Biology, University of Copenhagen, 2200 Copenhagen N, Denmark. [4] Department of Neuroscience, University of Copenhagen, 2200 Copenhagen N, Denmark. Correspondence and requests for materials should be addressed to K.D.R. (email: kasper.rand@sund.ku.dk) or to C.J.L. (email: cllo@sund.ku.dk)

Dopaminergic neurons control a broad array of basic psychological functions such as mood, motivation, reward, and addiction[1]. A major component in dopamine (DA) signaling homeostasis is the dopamine transporter (DAT)[2], which is responsible for terminating dopaminergic neurotransmission by rapid reuptake of DA. The central role of DAT in this process is substantiated by the fact that DAT malfunction is associated with a range of neurological diseases including early onset parkinsonism[3] and ADHD[4]. This makes DAT a major target for therapeutic drugs[5]. Furthermore, the rewarding and addictive effects of illicit drugs, such as cocaine and amphetamines, are also due to their inhibition of DAT function[6].

DAT belongs to the family of Neurotransmitter:Sodium Symporters (NSSs) together with transporters for serotonin (SERT) and norepinephrine (NET)[7]. NSSs are secondary active transporters that harness the energy stored in the Na$^+$ gradient to drive the uphill transport of their substrate across the cell membrane. X-ray structures of several NSS members across kingdoms have revealed highly similar structures[8–11], suggesting a preserved transport mechanism—even for transporters outside the NSS family[12].

Crystal structures of DAT from *Drosophila melanogaster* (dDAT), reveal a protein with 12 transmembrane domains (TMs) —with the first 10 TMs organized in an inverted symmetry between the first- and subsequent five TMs[10,13,14], also known as the LeuT-fold[8]. The binding site for substrate and ions are organized in the center of the protein through interactions with backbone and side chain residues in regions of TM 1, 3, 6, and 8. Notably, all current X-ray structures of eukaryotic NSS proteins[10,11,13–15] possess an outward-open conformation, which largely precludes interpretations on transporter function.

NSS proteins are believed to follow an alternating access transport mechanism[16,17] suggesting that the central binding site is alternatingly accessible to the intracellular and the extracellular side of the membrane. This requires the existence of external and internal 'gates', i.e., dynamic structural units that are capable of occluding access to the substrate binding site from the external- or internal environment, respectively. Crystal structures of bacterial homologs have been solved in multiple conformations, including outward-open[9], outward-occluded[8], and inward-open[9]. This allows for interpretations about transitions between the states. It has been speculated that alternating access is ensured through a rocking-bundle mechanism[18,19], where a core domain of TM 1, 2, 6, and 7 moves relative to a scaffold domain of TM 3, 4, 8, and 9. TMs 5 and 10 are linkers between the domains.

Most inferences on protein dynamics in NSS proteins have been derived from experiments on bacterial family members[8,9,20–27]. To gain specific insight into the molecular function of the more clinically relevant mammalian NSS proteins, a detailed understanding of the protein dynamics involved in the allosteric coupling between substrate- and Na$^+$-binding, and the control of the gating mechanisms, is essential.

Here we investigate how ion and substrate binding modulate the conformational dynamics of wild-type dDAT using hydrogen-deuterium exchange mass spectrometry (HDX-MS). Our results provide direct experimental insights into the conformational dynamics underlying substrate binding and isomerization between functional states of DAT. The results indicate that the alternating access mechanism in dDAT largely follows the rocking-bundle model with movements in the core domain relative to the scaffold domain. In addition, we find that several helical regions of dDAT undergo slow cooperative fluctuations that break multiple backbone hydrogen bonds resulting in an open and exchange-competent conformation, which refolds or closes at a rate that is significantly slower than the chemical exchange rate. These slow concerted fluctuations of helices are modulated by ligand binding, suggesting them to be an intrinsic part of the transport mechanism.

## Results

**MD simulations reveal structural stability of dDAT.** Despite the increasing attention to the role of the lipid bilayer in modulating NSS transport activity and function[28–32], studies of the structure and function of DAT is still generally limited to detergent micelles[10,13,14]. To address the possible risk of perturbing the structure and dynamics of DAT by the use of detergent micelles as membrane mimic, we carried out a comparison of all-atom MD simulations of dDAT embedded in a *n*-dodecyl β-D-maltoside (DDM) micelle and in a mixed lipid bilayer consisting of phosphatidylcholine (POPC), phosphatidylethanolamine (POPE), phosphatidylglycerol (POPG), and cholesterol (CHOL) in a 3:1:1:1 ratio. As existing structures of dDAT[10,13,14] contain truncations, deletions, and mutations, we first constructed a model of the wild-type protein using the outward-facing open crystal structure[13] (PDB ID: 4XP1) with two Na$^+$ ions, one Cl$^-$ ion and DA bound in the central binding pocket, and modeled the deleted part of the extracellular loop 2 (EL2) and corrected the mutations. We retained the crystallographically-resolved cholesterol molecule bordered between TM 1a, 5, and 7 as well as a cholesteryl hemisuccinate molecule (CHS; which we replaced by cholesterol) at the interface between TM 2, 7, and 11, which have been suggested to be important for transporter activity[13,32]. We performed two sets of 500 ns explicit-solvent all-atom MD simulations starting from our model of dDAT in a DDM micelle and in a POPC/POPE/POPG/CHOL bilayer (Fig. 1a). We found that the two Na$^+$ ions, the Cl$^-$ ion, DA and the cholesterol molecules were stable in their binding sites in the 500 ns MD simulations. Importantly, the root-mean-square deviations (RMSD) of the dDAT model from the starting dDAT crystal structure were below 0.25 nm (Fig. 1a) suggesting that the Na$^+$- and DA-bound state of dDAT in complex with cholesterol is stable in both the DDM micelle and the lipid bilayer. Moreover, the root-mean-square fluctuations (RMSF) of dDAT in the DDM micelle and the lipid bilayer were similar (Fig. 1b, c), indicating that the two systems display comparable structural dynamics in the simulations. The regions displaying the lowest RMSF are mainly located at the TM domains, while loop regions on the intracellular and extracellular sides show higher RMSF values. Importantly, the fluctuations for the two system setups are overall comparable. This suggests that dDAT embedded in a DDM micelle displays similar dynamics as in a POPC/POPE/POPG lipid bilayer indicating that the micelles are suitable membrane mimics for studies of the conformational dynamics of dDAT, where solubilization from the lipid cell membrane is a requirement. These results also confirm previous HDX-MS performed on the Xylose transporter[33] and the hydrophobic amino acid transporter LeuT[27] in DDM micelles and in nanodiscs, which also reported only minor differences in the conformational dynamics between the two settings. Taken together, although there are subtle differences in the structural stability between the MD simulations of dDAT in DDM micelles and a lipid bilayer, we find that our preparation of dDAT in DDM micelles as performed below, is a useful model system for probing dDAT conformational dynamics.

**Characterization of purified dDAT.** To investigate the conformational dynamics of DAT during ion and substrate binding, we expressed the wild-type dDAT in suspension HEK293 cells (Expi293F) using the BacMam expression system[34,35]. The transporter was purified by immobilized-metal affinity chromatography similarly to the procedure first described for the dDAT

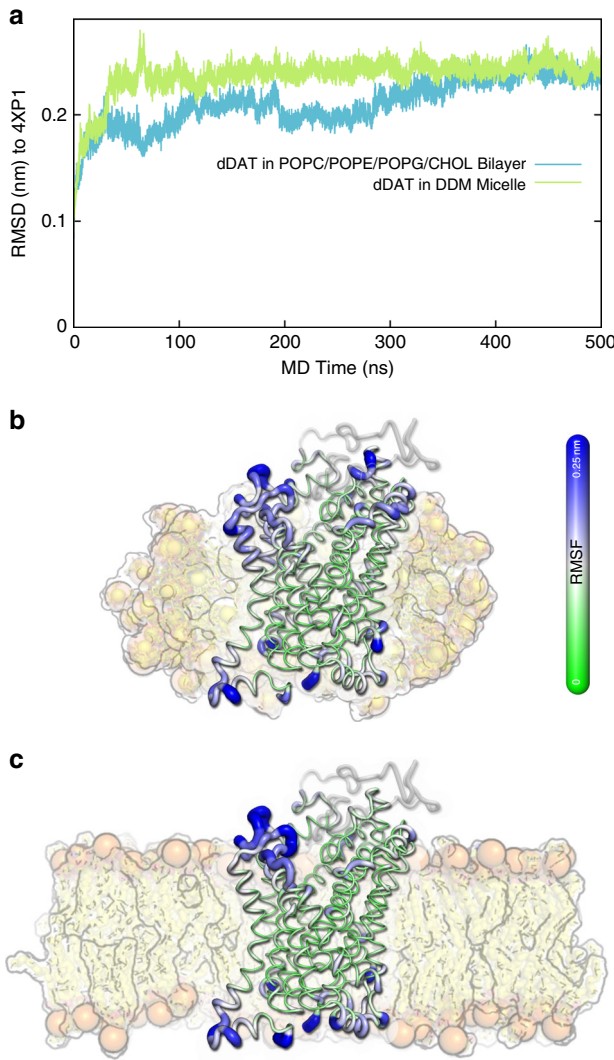

**Fig. 1** MD simulations of dDAT embedded in a detergent micelle and a lipid bilayer. **a** Trajectories of the root-mean-square deviation (RMSD) of the dDAT model to the dDAT crystal structure (PDB ID: 4XP1) when embedded in a DDM micelle (green) and in a POPC/POPE/POPG/CHOL bilayer (blue). **b**, **c** Structural fluctuations of dDAT in a DDM micelle (**b**) and a POPC/POPE/POPG/CHOL bilayer (**c**) are illustrated by projecting the corresponding root-mean-square fluctuation (RMSF) profiles onto the last frame of the MD trajectory of dDAT in the mixed bilayer. The thickness of the tube represents the amplitude of the fluctuations. To provide a better visualization, the thickness is scaled by the RMSF values in the region from 0 to 0.25 nm (the residues with RMSF > 0.25 nm are considered as the same as RMSF = 0.25 nm). The DDM micelle and the lipid bilayer used in the MD simulations are shown as background

crystallization constructs[10,13,14] in the presence of Na$^+$ in a buffer containing mixed micelles consisting of DDM, CHS, and lipids (POPC, POPE, and POPG). The presence of the cholesterol analog CHS and lipids in combination with DDM in the buffer increases the resemblance of the membrane mimic to a lipid bilayer and proved essential to preserve dDAT functionality throughout the purification procedure in agreement with previous reports[10,13]. The purity and activity of the purified transporter were evaluated by SDS-PAGE analysis and scintillation proximity assay (Supplementary Fig. 1). For subsequent binding and HDX-MS experiments, the transporter was left glycosylated to avoid possible deviation from its native conformational properties. Purified dDAT bound the high-affinity inhibitor

[3H]nisoxetine with a dissociation constant ($K_d$) of $80 ± 6$ nM (mean ± s.e.m., $n = 3$) in the presence of 200 mM NaCl. Based on the concentration of dDAT determined from SDS-PAGE analysis, the maximum number of binding sites ($B_{max}$) determined from [3H]nisoxetine saturation binding suggested that >95% of dDAT had preserved binding capability after detergent solubilization. Displacement of [3H]nisoxetine by DA yielded an inhibition constant ($K_i$) of $1.7 ± 0.2$ μM (mean ± s.e.m., $n = 3$). This is in agreement with the affinity of DA previously reported for wild-type dDAT expressed in transfected mammalian cells[13,36] indicating that a native functional conformation of the transporter is conserved in DDM/CHS/lipid mixed micelles.

**Dopamine transport intermediates**. To dissect the conformational dynamics of dDAT during DA transport, we studied intermediate states proposed by the alternating access model in solution using HDX-MS. The exchange rates of backbone amide hydrogens in a protein are sensitive reporters of the presence and relative stability of hydrogen bonds[37]. Hydrogens participating in bonding networks to form regions of higher-order structure, such as α-helices and β-sheets, will exchange according to the propensity of these hydrogen bonds to break as a result of dynamic protein motions. Accordingly, the exchange is correlated with conformational stability or flexibility of the protein. Thereby, HDX-MS also provides a means of measuring confined local changes in protein structure and dynamics, e.g., as a function of ion or ligand binding, within the entire protein[27,38–40].

In the absence of Na$^+$ and DA (apo state), we expect dDAT to fluctuate between multiple conformational intermediates in a dynamic equilibrium as has been observed for bacterial homologs[22,41]. Binding of ions and ligands will shift this equilibrium to favor distinct conformational states. Based on structural dynamics of bacterial homologs[22,41] using pairs of spin labels, we expect dDAT to be biased toward an outward-open conformation in the presence of Na$^+$, while the presence of Na$^+$ and DA combined would shift the equilibrium of sampled conformations towards an occluded and an inward-open state. Here we examined the HDX behavior of dDAT in an apo state, a Na$^+$-bound state and a Na$^+$- and DA-bound state from peptides covering 77.2% of the transporter (Fig. 2a; Supplementary Fig. 2a). Notably, only peptides showing signals with appropriate signal-to-noise ratio across all sampled states (Supplementary Fig. 2b) and time points (i.e., 0.25–480 min) were included in the HDX analysis (Supplementary Table 1, 2).

Comparing the HDX between the three states provided a direct spatially-resolved view of regions important for ion and substrate binding and for changes in isomerization between dDAT conformational states. The majority of identified dDAT peptides showed a gradual increase in deuterium uptake as a function of time indicating the presence of secondary structured regions, while the N- and C-termini as well as major parts of EL2 displayed maximum uptake after the first sampled time point (15 s) suggesting unstructured regions, which is in good agreement with the structural fold observed in the crystal structures[10,13,14] (Fig. 2b; Supplementary Fig. 3). Examples of deuterium uptake in specific DAT regions are shown in Fig. 2b. A large portion of the identified peptides showed changes in HDX upon ligand binding. The changes were mostly towards less deuterium uptake (see TM1a, TM6a, and TM8–9 in Fig. 2b) indicating a stabilizing effect, but increased uptake were also observed (see TM1b region in Fig. 2b). Other regions, on both the extra- and intracellular sides, showed no change in HDX upon ligand binding as seen for TM8 and the C-terminal region (Fig. 2b). The latter region is likely unstructured as it displayed maximum uptake after the first sampled time point.

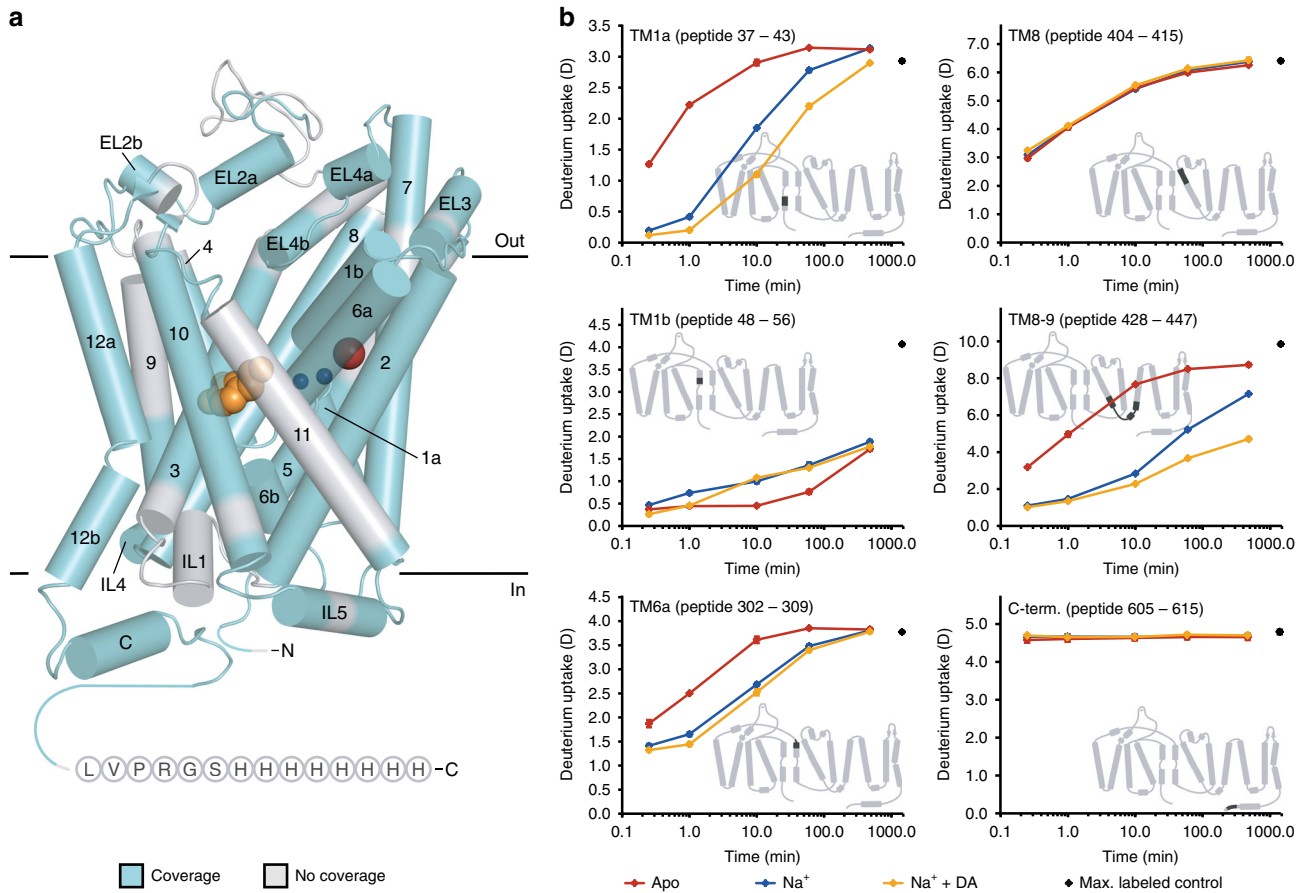

**Fig. 2** HDX of local regions of dDAT in different functional states. **a** Cylindrical representation of the crystal structure of dDAT in complex with two Na$^+$ ions (blue spheres), one Cl$^-$ ion (red sphere) and DA (orange spheres) (PDB ID: 4XP1). To obtain information about HDX of local regions of dDAT, we digested the transporter enzymatically and identified the generated peptides using MS. Regions of dDAT covered by identified peptides are highlighted in cyan. Wild-type dDAT was used for the HDX-MS experiments, however, the crystal structure shown has truncated N- and C-termini and deleted part of EL2. The C-terminal His-tag used for purification of the wild-type dDAT is added to the structure as circles with one-letter codes. **b** Deuterium uptake is plotted as a function of labeling time (i.e., 0.25–480 min) for representative peptides of dDAT. Red, blue, and orange curves illustrate the deuterium uptake for dDAT in the apo, Na$^+$-bound, and Na$^+$ + DA-bound state, respectively. Values represent means of three (apo and Na$^+$ + DA) or six (Na$^+$) independent measurements. Standard deviations are plotted as error bars but are in most instances too small to be visible. Maximum-labeled control samples are shown as black circles at 1440 min. Insets: The location of the corresponding peptide is highlighted in black on the topology map of dDAT. Source data are provided as a Source Data file

**Na$^+$-induced conformational dynamics**. To investigate the changes in conformational dynamics upon Na$^+$ binding, we compared the deuterium uptake over time between the apo state and the Na$^+$-bound state by calculating the difference in uptake for each individual dDAT peptide at all five sampled time points (Fig. 3a). Transition from the apo state to the ion-bound proposed outward-open state was predominantly characterized by a decrease in HDX corresponding to a decrease in dynamics and stabilization of H-bonding of regions both at the extracellular and intracellular side of dDAT (Fig. 3b, c). The effects were mainly focused around the bundle domain (i.e., TM1, 2, 6, and 7). Also, loop regions connecting the bundle domain were involved (i.e., EL1, EL3, EL4, and IL3). However, effects were also seen in the scaffold domain, where Na$^+$-induced stabilization was observed in the extracellular part of TM10 (residue 470–480) and the intracellular ends of TM8 and 9 connected by IL4 (residue 429–451). Also, a minor part of EL2 (residue 215–222), IL5 (residue 495–503) and the entire TM12 including the C-helix (residue 551–567, 569–594) showed significant stabilization upon Na$^+$ addition (Fig. 3).

Specifically, we observed reduced HDX of TM1a and the connected residues of the N-terminal end (residue 28–43), TM6b,

IL3 and the intracellular part of TM7 (residue 320–328, 330–347). These regions have all been proposed to participate in the opening and closing of the intracellular vestibule[9,22,24,42], and the observed stabilization suggests isomerization of the inner-gate to a closed conformation as expected upon Na$^+$-binding[43,44]. On the extracellular side, we observed increased HDX in the hinge region and TM1b (residue 45–56) together with large parts of TM7 (residue 349–357) and EL4 (residue 366–380) including the first helix of the loop (EL4a), while TM6a (residue 303–318) and TM2 (residue 58–73) showed decreased HDX upon ion binding. The domains of TM7 and EL4a have been associated with isomerization to the outward-open conformation[22], supporting the observations above on the intracellular side. Residues in the other domains are involved in the coordination of the two Na$^+$ and the Cl$^-$ in the crystal structures of dDAT[10,13,14]. Interestingly, the destabilizing and stabilizing effects seen in the bundle domain do not correlate with our observations from LeuT[27], where Na$^+$ stabilized the intracellular parts and destabilized the extracellular regions. Here, stabilization and destabilization do not show a clear pattern: the extracellular parts of TM1 and TM7 are destabilized, but stabilized in TM2 and TM6. A different pattern is seen on the intracellular side, where TM1 and TM6 are

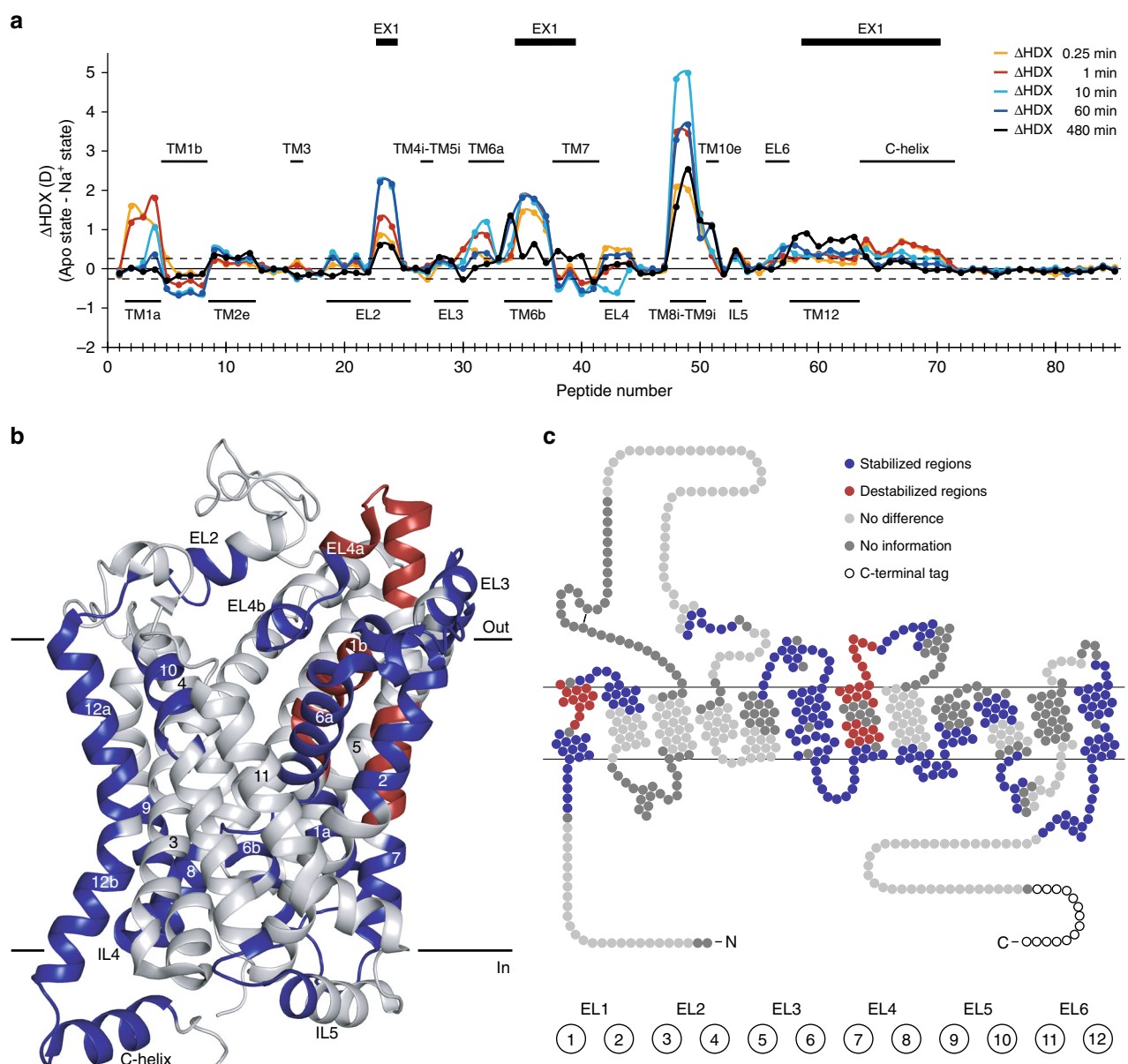

**Fig. 3** Effect on local HDX upon ion binding to dDAT. **a** Chart showing differences in the average deuterium uptake (ΔHDX) between the apo state and the Na$^+$-bound state for the 85 identified peptides at the five sampled time points (orange—0.25 min; red—1 min; cyan—10 min; blue—1 h; black—8 h). The individual dDAT peptides are arranged along the x-axis starting from the N-terminal and ending on the C-terminal. The peptide number refers to Supplementary Table 2. Positive and negative values indicate reduced and increased HDX, respectively, upon binding of Na$^+$. Values represent means of either three (apo state) or six (Na$^+$ state) independent measurements. Structural motifs in dDAT are marked along the x-axis together with regions showing correlated exchange kinetics (EX1) in at least one of the two states. The dotted lines (±0.26 D) mark a threshold value for significant differences in HDX corresponding to the 95% confidence interval, calculated from the pooled standard deviations for all time points. **b**, **c** Regions showing significant differences (Student's t-test p-value < 0.01) in deuterium uptake between the apo state and the Na$^+$-bound state for at least two consecutive time points are mapped onto the crystal structure (**b**) (PDB ID: 4XP1) and snake diagram (**c**) of dDAT. Regions are colored red and blue to indicate dDAT segments becoming destabilized (increased HDX) or stabilized (decreased HDX), respectively, upon binding of Na$^+$. Regions colored light grey displayed unchanged HDX while regions in dark gray were uncovered by peptide sequences. Wild-type dDAT regions including part of EL2 (residue 162–202) and the N- and C-termini (residue 1–24 and 601–645, respectively) are only marked on the snake diagram in **c** as they were not resolved in the crystal structure (**b**) or were truncated in the construct used for crystallization. Source data are provided as a Source Data file

stabilized, TM7 destabilized and no effect in TM2. The asymmetric conformational dynamics indicate that the bundle domain does not necessarily operate as a rigid bundle as suggested for LeuT[18,19], but rather mediate the transitions in the TM-domains that facilitate substrate binding and subsequent translocation. Within EL4 we also observed both destabilizing (EL4a, residue 366–380) and stabilizing (residue 381–390

including the second helix of EL4 (EL4b)) effects of ion binding. EL4 in LeuT exhibited a similar HDX profile between the outward-favoring condition and an inward-favoring mutant[23]. Together, this suggests that, as in LeuT, Na$^+$ binding favors the outward-open conformation of dDAT, with TM7 and EL4 participating in the opening of the extracellular vestibule. Na$^+$ - dependent decrease in HDX was especially pronounced in

the helical parts of EL2, and to a lesser extent in EL3 as well as in the extracellular part of TM10 (residue 470–480). This part of TM10 has Asp475 at its center, which likely closes access to the substrate binding site through a salt bridge to Arg52 in TM1b[8,45]. We see that Na$^+$ addition stabilizes the backbone interactions of the TM10 domain while destabilizing the H-bonding interactions of the TM1b backbone. Interestingly, the most noteworthy reduction in HDX upon Na$^+$ binding to dDAT was detected in TM8-IL4-TM9 (residue 429–451) suggesting a yet undefined role of this region in dDAT function.

TM3 (residue 112–132) and TM8 (residue 405–428) in the scaffold domain, which lines the central binding site and form interactions with ions in the crystal structures, did not show ion-induced changes in HDX. In addition, the intracellular part of TM5 (peptide 246–269) showed no change in conformational dynamics between the apo state and the Na$^+$-bound state indicating that dDAT under these conditions does not visit an inward-open conformation, where the intracellular part of TM5 changes the conformational dynamics. This is in contrast to what we observed for LeuT[27], and thus indicates a potentially important difference in the isomerization pattern between bacterial homologs and the eukaryotic dDAT.

**Dopamine-induced conformational dynamics**. To examine the conformational dynamics associated with substrate binding, we compared the HDX-MS profile of the Na$^+$ + DA-bound state to the Na$^+$-bound state (Fig. 4a). Overall, the observed differences are in the same regions as the differences between the apo state and the Na$^+$ - bound state (Supplementary Fig. 4), though not as pronounced.

Regions around the dopamine binding site in TM1 (residue 38–43, 45–56) and TM6 (residue 304–318, 326–328) revealed decreased dynamics, demonstrating a stabilization upon substrate binding (Fig. 4b, c). Notably, the regions involved in the extracellular salt bridge (TM1b (residue 45–56) and the extracellular part of TM10 (residue 470–480)) were stabilized further. The crystal structure of dDAT in complex with DA has confirmed several interactions between DA and residues in TM3 and TM8[8]; however, we do not observe altered dynamics in these helical segments. A reduction in deuterium uptake was observed for IL3, IL4, as well as for the intracellular parts of TM7, TM8, and TM9 (residue 330–347, 429–451). Thus, the Na$^+$-induced stabilization of intracellular regions involved in gate-closing was enhanced by binding of DA. The binding of DA also slightly stabilized the extracellular part of TM12 (TM12a, residue 556–567). The role of TM12 in eukaryotic NSSs is unclear; however, in the crystal structure of the human SERT, a CHS molecule was found to bind near TM12a[11], although this was not seen in the structures of dDAT.

DA did also induce destabilization in certain regions, particularly in the tip of EL4 and EL4b (residue 381–390) and the middle part of TM7 (residue 349–357) (Fig. 4). Effects were also seen in the N-terminal region connected to the beginning of TM1 (residue 28–37) and the extracellular part of TM2 (residue 58–65). Both TM1a and EL4 in LeuT have shown to be involved in transitioning to the inward facing state[27,46]. Changes in the intracellular part of TM7 have also been reported for leucine binding to LeuT, but there it caused a stabilizing effect[27]. The intracellular part of TM7 is part of the bundle domain. A destabilizing effect could indicate a transitioning towards an inward facing state.

Taken together, we find that the conformational dynamics associated with Na$^+$ and DA binding correlate with the rocking-bundle hypothesis illustrated with pronounced effects in TM1–2, 6–7 of the bundle domain relative to minor effects in TM3–5, 8–10 of the scaffold domain.

**Cooperative fluctuations in dDAT helical domains**. In an HDX-MS experiment, the exchange between hydrogens and deuteriums most commonly occurs through an EX2 time regime in which the exchanging protein segment undergoes opening and closing dynamics that occur at rates ($k_{op}$ and $k_{cl}$) that are significantly faster than the rate of the chemical exchange reaction $k_{ch}$ (specifically, $k_{cl} \gg k_{ch}$)[40,47]. Thus, the exchange of individual backbone amide hydrogens in this segment is uncorrelated and the observed rate of HDX ($k_{HDX}$) reports on the stability of local hydrogen-bonded structure in this segment. Such uncorrelated exchange appears as a gradually increasing binomially distributed isotopic envelope in the mass spectrum (see Supplementary Fig. 5a for example). While the majority of proteins exchange with EX2 kinetics under physiological conditions, there are increasing numbers of membrane proteins with reported EX1 kinetics[27,48,49]. In contrast to EX2 kinetics, EX1 kinetics is characterized by concerted opening and closing motions of several backbone amide hydrogen bonds that are slower than the rate of the chemical exchange (specifically, $k_{cl} \ll k_{ch}$). In the mass spectra, this is seen as a bimodal isotopic pattern resulting from the presence of a low-mass population, which has not yet undergone the cooperative fluctuating motion required for exchange, and a high-mass population in which the concerted opening event has occurred and multiple hydrogens have simultaneously exchanged for deuterium (Fig. 5, Supplementary Fig. 5b, 6). In the EX1 time regime, the observed rate of HDX is equal to the rate of the concerted opening ($k_{HDX} = k_{op}$)[40,47,50]. Thus, the presence of EX1 kinetics in a segment of a protein (e.g., a TM helix) allows a direct measure of the rate of conformational opening and the half-life of the closed/folded state ($t_{1/2}$). The low- and high-mass populations can be more or less separated on the $m/z$ axis depending on the number of involved backbone amide hydrogens, and the two populations might overlap due to the presence of EX2 and EX1 kinetics in the same peptide (EXX kinetics).

We detected EX1 and EXX kinetics in EL2 (residue 216–222), TM6b-IL3-TM7 (residue 330–347), the middle part of TM7 (residue 349–357), parts of TM12a (residue 562–567), TM12b (residue 569–577) as well as the helical part of the C-terminal (residue 578–594) in at least one of the investigated states (Fig. 5, Supplementary Fig. 5b, 6). Furthermore, signs of EX1/EXX kinetics were observed in TM8-IL4-TM9 (residue 429–451) and the extracellular part of TM10 (residue 470–480); however, a reliable quantitative bimodal deconvolution was not possible for peptides spanning these regions (referred to as 'potential EX1/ EXX kinetics (uncharacterized)' in Fig. 5). Interestingly, the transition from the low-mass population to the high-mass population was state-dependent, meaning that the opening rate ($k_{op}$) was modulated by Na$^+$ and DA binding, as seen for peptide 329–346 from the TM6b-IL3-TM7 region (Fig. 5c). Importantly, the existence of the EX1 exchange pattern was confirmed by being present in all replicates as well as overlapping peptides in all cases (Supplementary Fig. 5b). For some of the peptides, it was possible to obtain a quantitative measure for the rate of the conformational transition ($k_{op}$). For these peptides, we estimated the $k_{op}$ and thus the half-life of the closed (folded) state ($t_{1/2}$) by quantitating the time-resolved depletion of the low-mass population (Fig. 5d, Table 1). In general, $k_{op}$ for the regions showing correlated exchange was decreased upon Na$^+$ binding, and in some cases even further by the binding of DA (Table 1). The opening rate between different regions differed remarkably (from <0.001 s$^{-1}$ for peptide 577–593 to 0.02 s$^{-1}$ for peptide 329–347 in the apo state). For some regions the bimodal distribution fit was affected by the two populations not being sufficiently separated on the $m/z$ axis due to the presence of EXX. Such regions were not analyzed further for their rate of opening

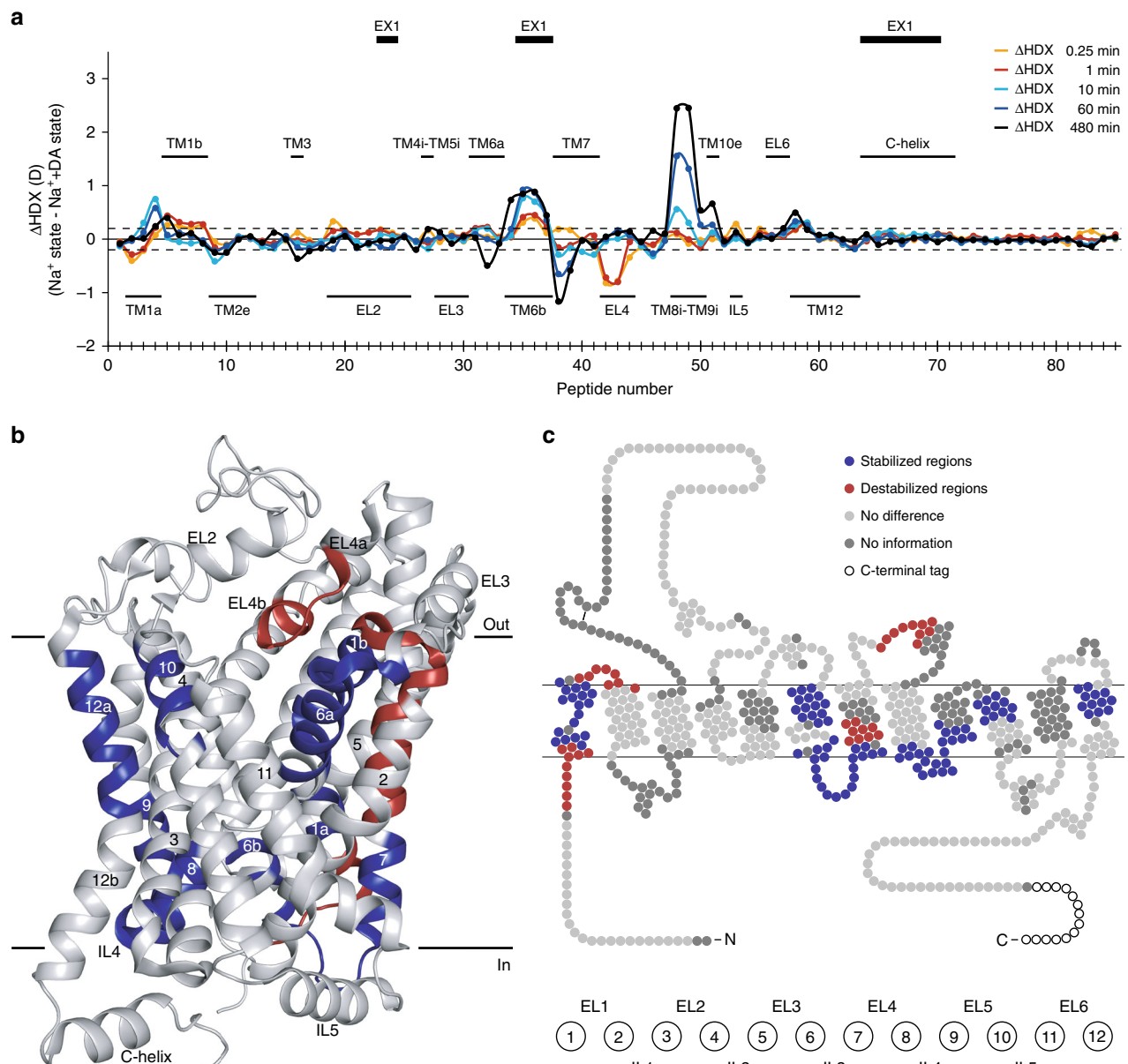

**Fig. 4** Effect on local HDX upon dopamine binding to dDAT. **a** Chart showing differences in the average deuterium uptake (ΔHDX) between the Na$^+$-bound state and the Na$^+$- and DA-bound state for the 85 identified peptides at the five sampled time points (orange—0.25 min; red—1 min; cyan—10 min; blue—1 h; black—8 h). The individual dDAT peptides are arranged along the x-axis starting from the N-terminal and ending on the C-terminal. The peptide number refers to Supplementary Table 2. Positive and negative values indicate reduced and increased HDX, respectively, upon binding of DA. Values represent means of either three (Na$^+$ + DA state) or six (Na$^+$ state) independent measurements. Structural motifs in dDAT are marked along the x-axis together with regions showing correlated exchange kinetics (EX1) in at least one of the two states. The dotted lines (±0.20 D) mark a threshold value for significant differences in HDX corresponding to the 95% confidence interval, calculated from the pooled standard deviations for all time points. **b**, **c** Regions showing significant differences (Student's t-test p-value < 0.01) in deuterium uptake between the Na$^+$-bound state and the Na$^+$- and DA-bound state for at least two consecutive time points are mapped onto the crystal structure (**b**) (PDB ID: 4XP1) and snake diagram (**c**) of dDAT. Regions are colored red and blue to indicate dDAT segments becoming destabilized (increased HDX) or stabilized (decreased HDX), respectively, upon binding of DA. Regions colored light gray displayed unchanged HDX while regions in dark gray were uncovered by peptide sequences. Wild-type dDAT regions including part of EL2 (residue 162–202) and the N- and C-termini (residue 1–24 and 601–645, respectively) are only marked on the snake diagram in **c** as they were not resolved in the crystal structure (**b**) or were truncated in the construct used for crystallization. Source data are provided as a Source Data file

(Table 1). Importantly, the difference between the high- and low-mass populations allowed us to estimate the number of amides participating in the fluctuation events (Table 1). These numbers reveal cooperative fluctuation of helical stretches spanning several residues. For peptide 215–222 covering the helical regions of EL2, the difference between the two populations corresponds to exchange of 4–5 backbone amide hydrogens, which reflects

cooperative fluctuation of more than one helical turn, while the difference seen for the TM6b-IL3-TM7 region corresponds to the exchange of 6–7 backbone amide hydrogens or involvement of 1.7–1.9 helical turns. The cooperative fluctuations in helical structures in dDAT reveal relatively slow local motion events. Notably, we did not sample longer time intervals than 480 min as we have evidence that the stability of the dDAT apo state at 25 °C

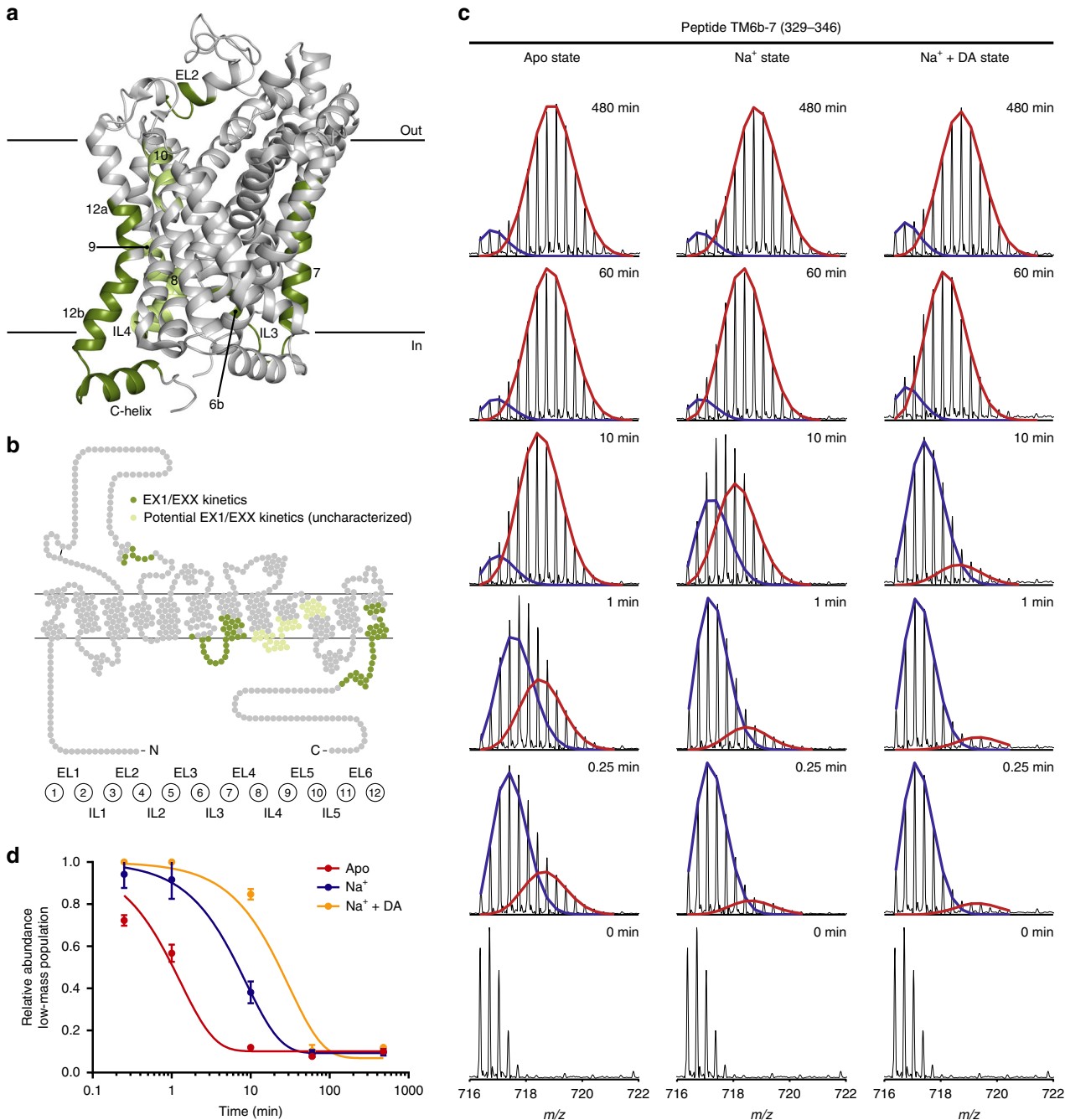

**Fig. 5** Correlated exchange kinetics of regions in dDAT. **a**, **b** Regions in dDAT, for which EX1 or EXX (a mixture of EX1 and EX2) kinetics were observed, are marked in dark green on the crystal structure (**a**) (PDB ID: 4XP1) and snake diagram (**b**) of dDAT. Signs of EX1/EXX kinetics were observed in regions marked in light green, but these could not be reliably analyzed. **c** Representative mass spectra for peptide 329–346, which covers TM6b-IL3-TM7, are shown for the three states at all sampled time points. Two binomial isotopic envelopes produced the best fit to the spectra—yielding a low- (blue) and high-mass (red) population. The rate of translation from the low-mass population to the high-mass population (i.e., $k_{op}$) was reduced upon binding of Na$^+$ relative to the apo state—and even more so by binding of Na$^+$ and DA combined. **d** The average relative abundance of the low-mass population is plotted against labeling time for the three states shown in **c**. Error bars indicate standard deviations ($n = 3$ for all time points for the apo state and the Na$^+$- and DA-bound state, $n = 6$ for all time points for the Na$^+$-bound state). The data were fitted to an exponential decay function. Data from overlapping peptides yielded similar results (Supplementary Fig. 5b, Table 1). Source data are provided as a Source Data file

would interfere with the analysis (Supplementary Fig. 7). For some regions showing correlated exchange kinetics, this did not allow a complete transition to the high-mass population, and thus kinetic parameters could not be extracted for these regions (TM12 and the middle part of TM7). Hereof, the stretches in TM12 (residue 561–567 and 569–576) showed very slow EX1 kinetics in the apo state, while ion and substrate resulted in

almost no exchange in the sampled time intervals. The middle part of TM7 (residue 349–357) also showed slow correlated exchange in the apo state, while the Na$^+$ and Na$^+$- and DA-bound states did not show EX1/EXX kinetics within the sampled time points. Finally, the C-helix (residue 578–594) showed correlated exchange for the apo state (Table 1) as well as the Na$^+$-bound and Na$^+$- and DA-bound states, however, extraction of

**Table 1 Kinetic parameters for dDAT regions showing EX1 or EXX kinetics**

| Peptide | State | $R^2$ | $k_{op}$ ($s^{-1}$) | $t_{1/2}$ (min) | #NHs |
|---|---|---|---|---|---|
| EL2 (215–222) | Apo | 0.86 | 0.016 [0.011; 0.024] | 0.73 [0.49; 1.10] | 4–5 |
| | Na+ | 0.99 | 0.0001 [0.0001; 0.0013] | 103.40 [89.11; 122.20] | |
| | Na+ + DA | 0.96 | 0.0002 [0.0001; 0.0003] | 56.86 [40.75; 80.91] | |
| TM6b-7 (329–346) | Apo | 0.95 | 0.01 [0.01; 0.02] | 0.89 [0.69; 1.17] | 5–6 |
| | Na+ | 0.98 | 0.0019 [0.0016; 0.0022] | 6.09 [5.22; 7.06] | |
| | Na+ + DA | 0.97 | 0.0005 [0.0004; 0.0007] | 21.14 [15.77; 27.86] | |
| TM6b-7 (329–347) | Apo | 0.81 | 0.02 [0.01; 0.03] | 0.65 [0.40; 1.02] | 6–7 |
| | Na+ | 0.99 | 0.002 [0.001; 0.002] | 7.12 [6.22; 8.15] | |
| | Na+ + DA | 0.99 | 0.0006 [0.0005; 0.0008] | 18.04 [14.42; 22.38] | |
| TM7 (348–357) | | | N.D. | | |
| TM8–9 (429–451) | | | N.D. | | |
| TM10 (469–480) | | | N.D. | | |
| TM12 (561–567) | | | N.D. | | |
| TM12 (569–576) | | | N.D. | | |
| C-helix (577–593) | Apo | 0.70 | 0.001 [0.0004; 0.0024] | 11.40 [4.77; 28.49] | 5–6 |
| | Na+ | | N.D. | | |
| | Na+ + DA | | N.D. | | |

The rate constant of opening ($k_{op}$) and the half-life of the low-mass population/folded state ($t_{1/2}$) were extracted from fitting the average relative abundance of the low-mass population as a function of labeling time to an exponential decay function for each state. ($n = 3$ for all time points for the apo state and the Na+-and DA-bound state, $n = 6$ for all time points for the Na+-bound state). Values defining the 95% confidence interval are given in square brackets. The coefficient of determination ($R^2$) is given to indicate the goodness of the fit. Furthermore, the number of amides (#NHs) participating in the cooperative fluctuation event was calculated as the difference in HDX between the high- and low-mass populations corrected for back-exchange. N.D. indicates peptides for which $k_{op}$ and $t_{1/2}$ could not be determined from the mass spectra (Supplementary Fig. 6). Source data are provided as a Source Data file

kinetic parameters was not possible for the two latter states due to spectral complication by the presence of EXX kinetics (Supplementary Fig. 6f). Nevertheless, it was clear that the opening rate was reduced by the presence of ion and substrate. Notably, we did not observe any bimodal distributed isotopic pattern in any of the sampled states for peptides covering TM1b, EL4 or the intracellular part of TM5 as previously observed for LeuT[23], suggesting differences in the mode of transition between the two. However, it is also possible that the differences are, at least in part, a consequence of sample conditions. To ensure sample stability of dDAT during the 8 h time-frame of the HDX-MS experiment, we performed analytical size-exclusion chromatography (SEC) of the apo state and the Na+-bound state after incubation at 25 °C in similar buffers as for the HDX experiments and matching time intervals (Supplementary Fig. 7). Analytical SEC showed that dDAT in Na+ is very stable and remains monomeric showing no formation of multimers over the 8 h period. We did observe that dDAT slowly formed multimers exclusively in the apo state with ~30% decrease in the monomeric peak after 8 h and concomitant increase in the oligomeric/void volume fraction. Importantly, our results do not indicate that the EX1 kinetics observed in both the apo, Na+-bound, and Na+ + DA-bound states are related to the multimerization observed for the apo state. First, the Na+-bound state of dDAT remains fully monomeric during the HDX-MS time-frame, yet we observe EX1 kinetics in both this and the Na+ + DA-bound state. Second, the EX1 kinetics observed in all regions of dDAT (apart from TM12) are much faster than the rate of multimerization observed by SEC, i.e., we have ~100% of interconversion to the high-mass fraction significantly before 8 h, e.g., TM6b-IL3-TM7 fully converts within 10 min in the apo state and most other regions even faster (Fig. 5, Table 1). Third, the EX1 kinetic behavior in

TM12 shows ~30% of the high-mass population after 8 h (Supplementary Fig. 6d, e), which could in principle correlate with the increased amount of multimerized dDAT. However, any precipitated dDAT resulting from multimerization would be removed during the filtration step before online pepsin cleavage and MS analysis. Furthermore, in order for irreversible multimerization to disguise itself as EX1 kinetics in TM12, it would then need to be due to a local irreversible unfolding of TM12 in a population of the protein over time. The remaining monomeric population would however likely exchange according to EX2. The low-mass population in TM12 does not detectably shift along the $m/z$ axis within the entire 8-h period, thus indicating that exchange observed in TM12 occurs exclusively via an EX1 mechanism.

## Discussion

Structural studies of NSS proteins have mainly focused on bacterial homologs such as LeuT and MhsT for which several crystal structures in different conformational states have been solved[8,9,25]. These structures have contributed with indispensable knowledge about the mechanism of action for this class of transport proteins. The addition of structures of dDAT[10] and hSERT[11] in a substrate- or inhibitor-bound outward-facing state has further advanced the studies to cover the eukaryotic transporters, which overall showed the same structural fold. However, the constructs used for the eukaryotic NSS proteins contained major truncations, deletions, and mutations that rendered most of them inactive in substrate transport. Furthermore, the information gained from crystal structures of eukaryotic NSS proteins needs further complementation by studies that capture the motions of the proteins in order to fully understand the

conformational rearrangements leading to substrate transport. So far, such studies include electron paramagnetic resonance (EPR) spectroscopy[22,44] and FRET-based strategies[20,21,51–53], which also require protein modifications and has so far only been applied to the bacterial homologues. Recent studies of LeuT[23,27] and SERT[54] have demonstrated the application of HDX-MS to assess the conformational dynamics of wild-type NSS proteins.

Here we intended to advance studies of the conformational dynamics of NSS proteins from the bacterial homologs to the eukaryotic dDAT in order to broaden the understanding of structural elements involved in the transport mechanism of DA. HDX-MS allowed us to obtain detailed insight into the conformational dynamics associated with both $Na^+$ and DA binding by identifying regions that changes stability and/or conformation upon ligand binding and, accordingly, must be implicated in the transport of substrate.

Our findings show that changes in conformational dynamics induced by ion and substrate binding mainly are located around the bundle domain (TM 1–2, 6–7), while the scaffold domain (TM 3–5, 8–10) overall remains unchanged with stabilization only localized to the extracellular part of TM10 and the end of helices continuing to loop regions (e.g., TM8-IL4-TM9). This is consistent with the rocking-bundle model for the alternating access transport mechanism, which predicts that isomerization from the outward-open to the inward-open conformation happens through a rotation of the bundle domain relative to the scaffold domain around an axis through the central substrate binding site[18,19]. However, the DA-induced increased dynamics in some parts of the bundle domain, in particular the middle part of TM7, indicates that the bundle domain moves as a more complex dynamic multi-jointed bundle rather than a rigid body. This is supported by studies of LeuT[8,9,22]. We also find that the addition of $Na^+$ results in the stabilization of regions forming the intracellular gate, such as regions including Arg27 in TM1, Tyr334 in IL3 and Asp435 in the intracellular part of TM8. Together with the destabilization of TM1b, TM7 and part of EL4 at the extracellular side, this is in agreement with the formation of an inward-closed/outward-open conformation of dDAT. The DA-induced decrease in dynamics of TM1b and TM6a, together with further stabilization of loop regions at the intracellular side (IL3, IL4), indicate stabilization of the extracellular as well as the intracellular gate, which would be expected in the transition to an outward-facing occluded conformation. Concurrent destabilization of the middle part of TM7, the hinge region and b-helix of EL4, part of the N-terminus and beginning of TM1a, suggests the importance of these regions in pushing dDAT towards an inward opening. Interestingly, the biggest effects of both ion and substrate binding were seen in the TM8-IL4-TM9 region. The loop region was highly dynamic in the apo state, but was stabilized in the presence of $Na^+$ and even further by DA. To our knowledge, this region has not been reported to be mechanistically important in structural studies of bacterial homologs. However, studies of the GABA transporter (GAT1) have demonstrated the role of IL4 as a regulatory barrier for substrate transport[55,56]. Furthermore, recent molecular dynamics simulations of homology modeled human DAT (hDAT) have reported the possible role of IL4 in triggering the inward opening and release of $Na^+$ through lipid mediated interaction with the N-terminal region[57,58]. Our recent HDX-MS data of LeuT importantly did not report any ion- or substrate-induced changes in the TM8-IL4-TM9 region[27]. Instead, we reported slow cooperative fluctuations (EX1) in the intracellular part of TM5, supporting the helical unwinding movement seen for this region in the crystal structure of MhsT[25], suggested to be required for $Na^+$ release. Our data on dDAT do not show the same slow cooperative fluctuations in this region, suggesting differences in the mode of transition, between the

eukaryotic dDAT and its prokaryotic cousins. While the temperature and pH was similar, we cannot rule out that sample preparation requirements for LeuT and dDAT could contribute to this observed difference. Indeed, the presence of lipids and CHS were absolutely essential for dDAT functionality and they will most likely contribute to the conformational dynamics of dDAT, possibly bringing it closer to its physiological state in a lipid bilayer. Clearly, the added lipids are coordinated at parts of dDAT that cannot be replaced by DDM in much the same way as cholesterol has its unique binding sites[10]. Nevertheless, cooperative fluctuations were seen in other helical regions of dDAT, which agreed well with HDX-MS data on LeuT (EL2, IL3, and TM7) reflecting on local similarities in the structural plasticity between the two. Like for LeuT, the ligand-dependent appearance and rate of opening during cooperative fluctuations suggest that these regions serve a role in the transport mechanism. The ligand-dependent correlated exchange seen for peptides covering TM12 and the C-helix mark a possible role for these regions in the transport mechanism or regulation of transporter activity. Recent studies of LeuT combining native MS and molecular dynamics simulations[59] have suggested a possible dimerization interface formed by lipid-mediated interaction between the intracellular part of TM9 and TM12. In contrast, studies on hDAT suggested that TM12 is not involved in dimer formation, possibly because of the kink in TM12[60]. The concomitant observation of EX1 kinetics in the C-helix support suggestions from hDAT of its involvement in activity-mediated regulation[6,61]. Slow cooperative fluctuations, or EX1 kinetics, are rare observations in proteins, but have also been observed in transporters such as P-glycoprotein[48], G-protein coupled receptors[49], and in enzymes[62]. EX1 behavior arises from correlated protein motions that occur slowly relative to the rate of the chemical exchange reaction. Our observation of EX1 in several TM helical regions of DAT is in good correlation with our earlier observations from LeuT[27] and can be interpreted as the hydrogen bonds of the helical segment breaking in a slow concerted manner, i.e., a partial unwinding or bending of the helical structure. This partially unwound helix stays open or unwound in a sufficient time-frame to allow several amides to exchange their hydrogens in a correlated manner (i.e., $k_{cl} \gg k_{ch}$) before closing again. Interestingly, the observation that the EX1 kinetics was attenuated by ligand binding could suggest that partial helix unwinding in the observed domains is an integrated part of the translocation mechanism. Another possibility is that it could be a result of local irreversible destabilization, especially because the most pronounced EX1 is observed in the apo state, which is the most unstable state. Indeed, our SEC shows that monomeric dDAT in the apo state is slowly converted into multimers and as a result, the monomeric fraction falls to ~66% after 8 h. However, dDAT in the $Na^+$-bound state exhibits no multimerization or change in stability even after 8 h, yet we still see pronounced EX1 kinetics in both the $Na^+$-bound and $Na^+$ + DA-bound states. Furthermore, for all domains exhibiting EX1 apart from TM12, the exchange to the fully deuterated high-mass isotopic envelope happens much faster ($t_{1/2} < 15$ min) than the partial formation of multimers observed exclusively for the apo state. Overall, this suggests that there is no correlation between the slow destabilization of the dDAT apo state and the EX1 kinetics observed across all DAT states. Notably, our apo state included $Cl^-$, which might obscure the observation of a correct apo state as binding of $Cl^-$ to dDAT might occur. Nevertheless, studies of DAT suggest a very low affinity for $Cl^-$ in the absence of $Na^+$[63].

Taken together, our HDX-MS data provide the first direct experimental evidence for spatially-resolved conformational dynamics of DAT. The findings support established models of the general alternating access transport mechanism based on studies

of bacterial homologs. Moreover, our data reveal substrate-induced conformational changes, which suggest key features for understanding the transport mechanism, structural plasticity, and regulation of eukaryotic NSSs. Comparison of our data with published HDX-MS studies of LeuT highlights important region-specific differences in dynamics between bacterial homologs and the eukaryotic transporters. Our work provides a comprehensive framework for future detailed studies into the transition path between transport states to fully elucidate the mechanism and regulation of clinically relevant NSSs.

## Methods

**Modeling and MD simulations**. A model of dDAT was constructed using the crystal structure of dDAT in complex with Na$^+$, Cl$^-$, and DA (PDB ID: 4XP1). In the crystal structure, the segment from Ser162 to Val202 is truncated for crystallization reasons, and was modeled using MODELLER9.18[64]. Similarly, mutations that had been introduced in the crystal structure were also reverted to the wild-type sequence. The two Na$^+$ ions, the Cl$^-$ ion, the DA molecule in the binding site were kept during model building together with the cholesterol molecule (CHOL) and the water molecules found in the crystal structure. Furthermore, the cholesterol analogue (cholesteryl hemisuccinate) found at the interface between TM2 and TM7 in the crystal structure (previously labelled[32] as site 2) was replaced by a CHOL molecule. The N-terminal (1–24) and C-terminal (601–645) regions, which are missing in the crystal structure, were not included in the model to decrease computational cost, and the termini were instead capped by an acetyl and amide group, respectively. Finally, the model was aligned to the dDAT structure (PDB ID: 4M48) in the Orientation of Protein in Membranes database[65]. DA in the binding site was modeled in its protonated state as suggested by Zeppelin et al.[32]. Force field parameters for the ligands were generated using the CHARMM General Force Field (CGenFF)[66] and the CHARMM-GUI web interface[67]. The CHARMM36m force field[68] was used for the protein.

To model dDAT in a mixed bilayer, the above model of dDAT was embedded into a flat, mixed lipid bilayer consisting of POPC/POPE/POPG/CHOL (3:1:1:1; with a total of 606 molecules) and solvated in a box of TIP3P water containing Na$^+$ and Cl$^-$ ions at 0.2 M. After equilibration, the size of the box was 13.7 nm, 13.7 nm and 12.2 nm in the $x$, $y$, and $z$ dimension, respectively. To mimic dDAT in a detergent micelle, the CHARMM-GUI web interface[67] was used to embed the model into a micelle consisting of 120 $n$-dodecyl β-D-maltoside (DDM) molecules and solvated in a box of TIP3P water molecules with a size of 11.8 nm in each dimension. The CHARMM36 lipid force field[69] was used for all lipid molecules.

In the MD simulations the temperature was kept constant at 300 K using a Nose-Hoover thermostat with a 1 ps coupling constant, and the pressure at 1.0 bar using the Parrinello-Rahman barostat with a 5 ps time coupling constant. A cutoff of 1.2 nm was applied for the van der Waals interactions using a switch function starting at 1.0 nm. The cutoff for the short-range electrostatic interactions was also at 1.2 nm and the long-range electrostatic interactions were calculated by means of the particle mesh Ewald decomposition algorithm with a 0.12 nm mesh spacing. All MD simulations were performed using Gromacs2018[70]. One 500 ns and four 100 ns simulations were performed for each system of dDAT in the lipid bilayer and the micelle.

**Construct**. Full-length dDAT with a C-terminal thrombin site (LVPRGS) followed by an 8 histidine-tag and flanking unique restriction sites (BssHII–MluI–EcoRI–ApaI–dDAT–AgeI–SalI–NotI–XbaI) was synthesized by GenScript Inc. (Piscataway, NJ) and cloned into the pEG BacMam expression vector[35] using BssHII and XbaI by GenScript Inc. (Piscataway, NJ).

**Expression and purification**. The wild-type dDAT was expressed by transduction of mammalian Expi293F cells (Gibco) grown in suspension with baculovirus produced in Sf9 cells (Expression Systems). Cells were harvested by centrifugation and membranes were prepared in 20 mM Tris, pH 8.0, 150 mM NaCl, 30% glycerol, 10 μg ml$^{-1}$ benzamidine and 10 μg ml$^{-1}$ leupeptin by dounce homogenization followed by sonication. Unlysed cells and cell debris were removed by centrifugation at 750 × $g$ and membranes were then pelleted at 125,000 × $g$ for 3 h. Membranes were further solubilized in buffer containing 40 mM Tris, pH 8.0, 150 mM NaCl, 20 mM $n$-dodecyl β-D-maltoside (DDM), 4 mM cholesteryl hemisuccinate (CHS), 5 μg ml$^{-1}$ benzamidine and 10 μg ml$^{-1}$ leupeptin for 1.5 h by gentle end over end mixing followed by centrifugation at 55,000 × $g$ for 30 min. The clarified supernatant was batch incubated with His-Pur Ni-NTA resin (Thermo Fisher Scientific) and 25 mM imidazole for 2 h at 4 °C under gentle end over end mixing. The resin was washed in buffer A (40 mM Tris, pH 8.0, 5% glycerol, 14 μM lipids (1-palmitoyl-2-oleoyl-$sn$-glycero-3-phosphocoline (POPC), 1-palmitoyl-2-oleoyl-$sn$-glycero-3-phosphoethanolamine (POPE), and 1-palmitoyl-2-oleoyl-$sn$-glycero-3-phospho-(1'-rac-glycerol) (POPG) at a weight ratio of 3:1:1), 1 mM DDM, 0.2 mM CHS) containing 300 mM NaCl, 5 μg ml$^{-1}$ benzamidine and 10 μg ml$^{-1}$ leupeptin with a stepwise gradient of 6 column volumes of buffer supplemented with 30 mM imidazole followed by 6 column volumes of 60 mM imidazole buffer, and the protein was then eluted with 300 mM imidazole

buffer. Fractions containing dDAT was pooled and stored at −80 °C until further use. Buffers and protein samples were kept on ice or at 4 °C at all times during the purification. Purified dDAT for analytical size-exclusion chromatography and HDX-MS experiments was thawed and dialyzed two times for 2 h followed by one time overnight at 4 °C against buffer A containing 200 mM of either NaCl or CsCl using 0.5 ml Slide-A-Lyzer MINI dialysis devices (10 kDa MWCO, Thermo Fisher Scientific).

**SDS-PAGE**. The purity and concentration of purified dDAT were analyzed by SDS-PAGE. Deglycosylated dDAT was prepared by incubation with PNGaseF (Sigma–Aldrich) overnight at 25 °C. Samples were mixed with 4X Laemmli sample buffer containing 167 mM DTT and separated on Any kD Mini-PROTEAN TGX Precast gels (BioRad). Gels were stained by GelCode Blue Stain reagent (Thermo Fisher Scientific) according to the protocol provided by the manufacturer. Bovine-serum albumin (Sigma–Aldrich) standards were used for quantification of dDAT. Gels were analyzed and quantified using the GelAnalyzer 2010 software.

**Ligand binding assay**. The activity of purified dDAT was assessed by scintillation proximity assay (SPA). 50 ng per well (7 nM) of protein was incubated with 1.25 mg ml$^{-1}$ copper yttrium silicate (Cu-YSi) beads (PerkinElmer) in buffer A supplemented with 200 mM NaCl. Nisoxetine saturation binding was set up using 10% [$^3$H]nisoxetine (79.8 Ci mmol$^{-1}$; PerkinElmer). DA competition binding assays was performed with 30 nM [$^3$H]nisoxetine and increasing concentration of unlabeled DA. Non-specific binding was determined in the presence of 100 μM unlabelled nortriptyline. SPA experiments were set up in triplicate wells of a 96-well white solid clear bottom plate. The samples were incubated for 30 min at room temperature. The samples were further incubated for 16 h incubation at 4 °C for nisoxetine saturation binding experiments. [$^3$H]nisoxetine binding was monitored using a MicroBeta liquid scintillation plate counter (PerkinElmer) using a 1-min counting protocol. Data were analyzed by non-linear regression analysis and fitted to a one-site saturation or dose-response function, respectively, using GraphPad Prism 7 software (GraphPad, San Diego, CA).

**Analytical size-exclusion chromatography**. dDAT samples were incubated at 25 °C in buffer A containing either 200 mM NaCl or CsCl. After various time intervals (0, 1, 4, and 8 h) the samples were subjected to size-exclusion chromatography on a Superose 6 Increase 10/300 GL column (GE Healthcare) pre-equilibrated with the corresponding buffer A with 200 mM NaCl or CsCl.

**Hydrogen-deuterium exchange**. Prior to HDX, dDAT was incubated at 25 °C for 30 min at a concentration of 1.5 μM in buffer A containing 200 mM of either NaCl, CsCl, or 200 mM NaCl and 400 μM DA. The exchange reaction was initiated by diluting the dDAT samples 1:4 with 94% D$_2$O buffer A supplemented with 200 mM of either NaCl or CsCl at 25 °C resulting in 75.2% deuterium content during HDX and 40 μM DA in the NaCl and DA condition. The labeling reaction was quenched at various time points (15 s, 1 min, 10 min, 60 min, and 480 min) by mixing a sample 1:1 with ice cold 220 mM phosphate buffer, pH 2.3, 2 M urea. Quenched samples were immediately frozen and stored at −80 °C until further use. For maximum-labeled control samples, dDAT was predigested using the same conditions as for the HDX experiments. After online pepsin digestion, dDAT peptides were desalted, eluted, and collected manually in a single fraction and lyophilized. The dDAT peptides were then reconstituted in 75.2% deuterated buffer A containing 200 mM NaCl and incubated for 24 h at 25 °C to reach full deuteration before quenching the reaction. The HDX-MS data presented in this paper is the average of at least three technical replicate measurements for each time point.

**Liquid chromatography and mass spectrometry**. Quenched samples (15 pmol) were thawed and loaded onto a cooled (0 °C) nanoACQUITY UPLC system (Waters) with HDX technology. Samples were first subjected to proteolytic digestion at 20 °C on an in-house packed pepsin column containing immobilized pepsin agarose resin (Thermo Fisher scientific) followed by rapid desalting on a C8 trap column (VanGuard pre-column ACQUITY UPLC BEH C8 1.7 μm, Waters) for 3 min at a flow rate of 200 μl min$^{-1}$ solvent A (0.23% formic acid in water, pH 2.5). The peptides were then separated by reversed-phase chromatography over a C8 analytical column (ACQUITY UPLC BEH C8 1.7 μm, 100 mm, Waters) with a C8 trap column in front using a linear gradient from 8–30% solvent B (0.23% formic acid in acetonitrile) over 10 min at a flow rate of 40 μl min$^{-1}$. Subsequently, peptides were analyzed on a hybrid Q-TOF SYNAPT G2-Si mass spectrometer (Waters) with electrospray ionization in positive ion mode and lock-mass correction using Glu-fibrinopeptide B. Ion mobility was used to further separate ions in the gas-phase in order to enhance peak capacity and minimize spectral overlap. The ion mobility cell was operated using a constant nitrogen flow of 90 ml min$^{-1}$ at a wave velocity of 580 m s$^{-1}$ and a wave height of 40 V. The maximum-labeled control samples were processed and analyzed using the HDX-MS workflow as described above with the only exception that the pepsin column was removed to avoid additional digestion. Non-deuterated samples were used for peptide identification by tandem mass spectrometry (CID) using a combination of data-independent acquisition (DIA) and data-dependent acquisition (DDA) with the ion mobility cell turned off.

**HDX-MS data evaluation and statistical analysis**. Peptides were identified from non-deuterated samples using ProteinLynx Global Server 3.0 (PLGS) (Waters). Only peptide hits with a PLGS Ladder Score above 1.0 and with a mass error below 10 ppm for the precursor ion were accepted for peptides identified by DDA. Peptides identified by DIA were filtered using DynamX 3.0 (Waters) so that only peptides with at least two fragmentation products, 0.2 fragment ions per amino acid, and below 10 ppm mass error for the precursor ion were accepted. Furthermore, peptides had to be identified in 50% of the DIA runs. The deuterium uptake was determined for all identified peptides using DynamX 3.0 (Waters). All peptide assignments were manually verified and noisy and overlapping spectral data were discarded from the HDX analysis. Maximum-labeled control samples were used to calculate the back-exchange (BE) for all individual peptides by the equation

$$\mathrm{BE}\,(\%) = \left(1 - \frac{m_{75.2\%} - m_{0\%}}{m_{\mathrm{MAX}} - m_{0\%}}\right) \cdot 100\% \tag{1}$$

where $m_{75.2\%}$ is the mass of the maximum-labeled peptide, $m_{0\%}$ is the mass of the non-deuterated peptide, and $m_{\mathrm{MAX}}$ is the theoretical maximum deuterium uptake of the peptide (excluding N-terminus and prolines). The calculated back-exchange was used to normalize the back-exchange between measuring days allowing quantitative comparison of samples not measured on the same day. HX-Express 2.0 was used to further analyze peptides showing bimodal isotopic patterns upon deuteration. The number of backbone amide hydrogens undergoing correlated exchange (#NHs) was determined using the equation

$$\#\mathrm{NHs} = \frac{\Delta\mathrm{HDX} \cdot 100\%}{100\% - \mathrm{BE}} \tag{2}$$

where $\Delta$HDX is the difference in HDX between the low- and high-mass population, which was calculated ($n = 3$) and corrected according to the measured back-exchange (BE) of the respective peptide. For peptides displaying EX1 kinetics, the relative abundance of the two isotopic envelopes were determined using HX-Express 2.0, and the relative abundance of the low-mass population ($n = 3$) was plotted against the labeling time. The data were then fitted to an exponential decay function

$$y = \mathrm{e}^{-k_{\mathrm{op}} \cdot x} \tag{3}$$

using GraphPad Prism 7 software (GraphPad, San Diego, CA) to extract the best-fit values for the kinetic parameters (i.e., $k_{\mathrm{op}}$ and the half-life of the folded state). Comparisons of states for all identified peptides were performed in Excels software (Microsoft) using either a homoscedastic or a heteoscedastic Student's $t$-test ($\alpha = 0.01$) depending on a F-test ($\alpha = 0.05$), which compared the variance of deuterium uptake from two different states for each single peptide at a single time point. Peptides were only considered to have a significant difference in HDX between two states if two consecutive time points showed a significant difference in deuterium incorporation ($p < 0.01$). A significance threshold for all peptides used to define a significant difference in the difference charts was set to 0.26 Da, 0.20 Da, or 0.23 Da for differences in HDX between the $Cs^+$ and $Na^+$ states, the $Na^+$ and $Na^+ + DA$ states, or $Cs^+$ and $Na^+ + DA$ states, respectively, corresponding to the 95% confidence interval (CI) calculated according to

$$\mathrm{CI} = \bar{x} \pm t \cdot \frac{\sigma}{\sqrt{n}} \tag{4}$$

Here, $\bar{x}$ is the average difference in deuterium content assuming a zero-centered distribution ($\bar{x} = 0$), $t$ is 4.303 for the 95% CI with 2 degrees of freedom, $\sigma$ is the pooled propagated standard deviation of differences in deuterium content for all peptides across all time points (i.e., 0.25–480 min) for the two states compared, and $n$ is the number of replicate samples ($n = 3$). HDX results were mapped onto the dDAT crystal structure (PDB ID: 4XP1) using PyMOL.

To allow access to the HDX data of this study, the HDX Summary Table (Supplementary Table 1) and the HDX Data Table (Supplementary Table 2) are included in the Supplementary Information according to the community-based recommendations[71].

**Reporting summary**. Further information on research design is available in the Nature Research Reporting Summary linked to this article.

## Data availability

Data supporting the findings of this manuscript are available from the corresponding authors upon reasonable request. A reporting summary for this Article is available as a Supplementary Information file. The source data underlying Figs. 2, 3, 4, 5d, Table 1 and Supplementary Figs. 1, 3, 4, 7 are provided as a Source Data file. Mass spectrometry data files including the processed DynamX file and an overview of the HDX-MS data (Supplementary Tables 1 and 2) have been deposited to the PRoteomics IDEntification (PRIDE) Database with the dataset identifier PXD013841.

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

## Acknowledgements

The pEG BacMam vector was kindly provided by Dr. Eric Gouaux, Vollum Institute, Oregon Health & Science University. We thank Giuseppe Cazzamali and the rest of the staff at the Protein Production and Characterization Platform at the Novo Nordisk Foundation Center for Protein Research, University of Copenhagen, for help with protein expression, cell work, and for making it possible to perform this part of the work at their facility. Also great thanks to Lei Shi, Investigator at NIDA, NIH for helpful suggestions. The work was supported in part by the Danish Council for Independent Research (0602-02100B and 4183-00581 to C.J.L. and 0602-02740B to K.D.R.), the Novo Nordisk Foundation (NNF17OC0028582 to C.J.L.), the UCPH bioSYNergy center of excellence (C.J.L.), the Lundbeck Foundation BRAINSTRUC initiative (K.L.-L.), and the Lundbeck Foundation (S.G.F.R.).

## Author contributions

A.K.N. performed the experiments and data analysis with assistance from I.R.M., S.G.F.R., and K.D.R. Y.W. and K.L.-L. constructed the model of dDAT and carried out the MD simulations. A.K.N., K.D.R., and C.J.L. designed the experiments and interpreted the data with support from I.R.M., S.G.F.R., Y.W., and K.L.-L. A.K.N., K.D.R., and C.J.L. prepared the manuscript and all the authors commented.

## Additional information

**Competing interests:** The authors declare no competing interests.

