## [Peer Review File · Nature Communications]

Reviewers' Comments:

Reviewer #1:

Remarks to the Author:

Nielsen et al. & Rand, Gotfryd present a HDX-MS study of the *Drosophila* dopamine transporter dDAT. They find local dynamics induced by Na⁺ and dopamine binding and propose also slow long-range, ligand-dependent and cooperative fluctuations. The study presents good data and fine work from a well-purified sample (although to make a bold claim that this is perhaps the first report on wild-type DAT purification appears excessive).

Overall, the study however remains preliminary, descriptive and confirmative, and the HDX-MS coverage remains limited and prevent the analysis to reach a complete proper picture of cooperative effects and causation

Pepsin is used for cleavage – could other proteases complement it? Furthermore, the study is performed in detergent, not in a membraneous environment, so subtle details of dynamics and cooperative movements are very likely flawed by the detergent-solubilized state of the protein. Line 326 attempts a justification of detergent-to-lipid bilayer analogy by comparing it to LeuT studies in either lipid or detergent – this is not a valid comparison and validation of the current study.

The results invite a much wider use of MD simulations (past literature and new studies/analyses) to reach further qualification and mechanistic insight from the models of dynamics proposed and how they affect the transport activity overall.

Line 56: "...first insights into the conformational dynamics underlying substrate binding and their possible role....". This is not a fair justification and qualification of the study given the enormous literature on NSS transport and dynamics

Generally the manuscript is difficult to follow, and needs a substantial rewriting and mechanistic focus.

Reviewer #2:

Remarks to the Author:

In this article, the authors investigate the conformational dynamics of the human dopamine transporter DAT. They use HDX-MS in three different conditions (apo, with Na⁺, with Na⁺ and substrate dopamine) to identify the regions involved in the structural rearrangements underpinning the transport cycle. Their findings seem to overall confirm the rocking-bundle alternating-access mechanism despite some discrepancies. They identify a region hugely stabilized upon ligand binding which was not previously pointed out as mechanistically relevant.

Overall, it is a well-written manuscript, with a solid experimental set-up, which gives novel insights into the mechanism of an important drug target and will lay the foundation for other structural studies.

Major concerns

1. My main concern is whether the EX1 kinetics observed might be indicative of a local unfolding that is not physiologically relevant but it may be just caused by the protein degrading over time during the HDX experiment which is conducted at 25C. This should be tested and reported in the manuscript. The authors could consider conducting some parallel experiments (e.g. FTIR, CD or analytical SEC) to assess the stability of the transporter over time. In line 230 the authors suggest they actually have some data on stability but they don't elaborate further – they should show this data. Since eukaryotic transporters are notoriously unstable, such control is essential. Most of this EX1 kinetics were observed in the apo state, which I believe is the most unstable.

2. I find the results section difficult to read and partially confusing. Moreover, the figures could be improved, e.g in figure 1, each highlighted peptide could be represented individually on a cartoon version of dDAT and added as an inset in the uptake plots. Please annotate the topological maps with helix numbers, and indicate where the intracellular and extracellular regions are.

3. Could the authors explain why they add lipids in the buffers? Is it common practice for dDAT purification? The presence of lipids is not trivial because it has been shown to modulate the conformational dynamics of other transporters and receptors, even in the absence of a bilayer (doi: 10.1038/nchembio.1960, doi: 10.1038/nsmb.3262). The authors might want to discuss this possibility with respect to their findings.

Other comments

The use the word "novel" twice in the same sentence – one time must be enough.

Line 70: Could the authors elaborate a bit on the optimization of their purification protocol? Is the use of mixed micelles of detergent and lipids a common practice for the purification of mammalian transporters?

Fig.1: What is the meaning of the color gradient of the cyan peptides? This seems a bit confusing.

Line: 140 "Interestingly, the destabilizing effects in TM1b and TM7 contrast the stabilizing effects in TM6a and TM2." Could the authors elaborate on this observation? Why is it interesting? Is it interesting because it doesn't fit with what the authors expected from an outward-open conformation?

Line 142: "Similar effects were seen for EL4 in LeuT when comparing the outward-favoring condition with an inward-favoring LeuT mutant, indicating that ion binding favors the outward-open conformation of dDAT, with TM7 and EL4 participating in the opening of the extracellular vestibule". I find this sentence confusing: how do LeuT results support the fact that dDAT is in an outward open conformation upon sodium binding?

Figure 2: the panel b is very difficult to read. Maybe the authors could show only stabilized peptides on one structure and only destabilized on another structure? And colour-coded the peptide ID as well?

Fig.3. Same as above. Panel b is very hard to read and panel c should be annotated.

Discussion: The discussion pretty much suggests that at least part of the results will be very different in a lipid bilayer. Have the authors carried out MD simulations to provide some hints on how much different things could be?

Supplementary

- Extended data figure 2. Not sure I understand the purpose of panel b. The representative spectra are shown at only one time point. This doesn't match the legend.
- Thanks for providing the uptake plots, that's very useful.

Reviewer #3:

Remarks to the Author:

The manuscript by Nielsen et al. seeks to map the conformational dynamics of the dopamine transporter by substrate and sodium ions. This is an interesting manuscript and exciting study that provides insights into the functioning of the neurotransmitter: sodium symporters. However, there are several questions that I have that relate to the methodology and interpretation that needs to be better explained in a revision before the manuscript can be accepted for publication. These are highlighted below:

Major points:

- 1) My concern is that the dopamine transporter in this study is detergent solubilized and even though it is examined in a mixture of micelles, it is still not embedded in a lipid matrix. Given this, it is unclear how it can be concluded that "Our HDX-MS data provides the first spatially-resolved view of the conformational dynamics of DAT". The authors should more clearly describe how anchoring in lipid might alter/enhance function of the transporter.
- 2) Dopamine transport is directional. It is unclear how a mixture of inside-out and outside-in orientation of transporters in detergent micelles, can allow correlation of dynamics as read out by HDX with function?
- 3) In figure 1, the Y-axis should be average uptake of deuterons. Not clear what is meant by Relative Uptake? Relative to what? It seems like absolute exchange.
- 4) According to the authors as stated in the abstract, "slow cooperative fluctuations are observable in several domains of the transporter, which could be a novel molecular mechanism that assists in the transporter function" I am not clear how these slow fluctuations contribute to transporter function.
- 5) The K_m of dopamine is 8.2 μM according to reference 13 (Wang et al. (2013)), but the K_i here reported is 1.7 μM . The authors need to explain what nisoxetine is. The papers cited are measuring dopamine transport and so are reporting K_m values but here the authors are reporting displacement. The authors need to clarify.
- 6) On page 6, line 104, it is not clear what is meant by "the majority of identified dDAT peptides showed a gradual increase in deuterium uptake as a function of time, indicating the presence of secondary structured regions in good agreement with the structural folds observed in the crystal structures" How is a gradual increase in deuterium uptake as a function of time correlated with secondary structured regions? How is this being inferred?
- 7) In figure 1, the red chlorine atom and the second sodium ion are not evident. The cyan gradation is not explained. TM1b for instance is a darker shade of cyan. How much a difference separates the dark cyan from light cyan? It is also not useful to highlight peptides showing difference in uptake between at least two states. What time points are being considered. Is it the maximal difference at any timepoint that it may be at? For instance, for peptide 320-328 (TM6 hinge), are you representing a difference of 3.0 D (for apo) minus 0.5 D (for $\text{Na}^+ + \text{DA}$) at 5000 min as your dark cyan. How is this dark cyan equivalent to peptide 48-56 (TM1b) where $\text{Na}^+ + \text{DA}$ shows greater exchange than apo. These two peptides show opposing effects of sodium and DA and so the cyan coloration of the structure is misleading and uninformative!
- 8) The entire results section on the Cooperative fluctuations in dDAT helical domains needs to be revised. It is more of an observation. It is not clear how sodium contributes to these cooperative fluctuations and how these might contribute to transporter function.
- 9) In Extended figure 1, why was dDAT deglycosylated? When it is stated that the glycosylation was retained on the protein for effective function (line 73).
- 10) What is meant by 'POTENTIAL EX1/EXX kinetics' in Figure 4?
- 11) The parameters for ion mobility separations (line 400) or a reference should be provided.
- 12) How was the mix of DDM + cholesteryl hemisuccinate (CHS) and lipids (POPC, POPE and POPG) derived?

Reviewers' comments:

Reviewer #1 (Remarks to the Author):

Nielsen et al. & Rand, Gotfryd present a HDX-MS study of the *Drosophila* dopamine transporter dDAT. They find local dynamics induced by Na⁺ and dopamine binding and propose also slow long-range, ligand-dependent and cooperative fluctuations. The study presents good data and fine work from a well-purified sample (although to make a bold claim that this is perhaps the first report on wild-type DAT purification appears excessive).

Our response:

We thank the reviewer for complementing our work and the quality of the data presented. It is, to the best of our knowledge, the first report on purified wild-type DAT. The previous reports – all from Gouaux lab – are x-ray crystallographic studies performed on thermostabilized mutants (Penmatsa et al. 2013, Nature; Penmatsa et al. 2015, NSMB; Wang et al. 2015, Nature; Navratna et al. 2018, PLoS One). If the reviewer is aware of any other published reports, we will highly appreciate the information.

Comment #1:

Overall, the study however remains preliminary, descriptive and confirmative, and the HDX-MS coverage remains limited and prevent the analysis to reach a complete proper picture of cooperative effects and causation

Our response:

We are confused by the comment that our study should be preliminary. All our HDX-MS data presented in the paper is the average of at least three technical replicates and thus includes robust statistical analysis of significance (see Methods section). We agree that some of our experimental data confirm what has been proposed previously by computational analysis, and furthermore the data also adds important new insights to the proposed transport mechanism. We would have preferred to have obtained a sequence coverage of 100%, however, this is practically impossible for membrane proteins of this size with 12 membrane spanning hydrophobic domains. Please note that our coverage reflects the effective sequence coverage, hence, only peptides displaying signals across all investigated states, time points, and replicates are included. Our coverage of 77% is higher than what is usually reported for HDX-MS experiments of complex integral membrane proteins and is even comparable to what can sometimes be achieved for HDX-MS experiments of soluble. For comparison, we achieved an effective sequence coverage of 71% in our previous paper on LeuT (Merkle et al. 2018, Sci Adv.). Adikary et al. (2017, PNAS) reported in their study on LeuT an effective sequence coverage of 27%, and Canul-Tec et al. (2016, Nature) reported an effective coverage for EAAT-1 of 76%. We are convinced that the fact that we are able to assess and compare dynamics within 77% of the protein across all examined states, makes our HDX-MS results superior to conventional biochemical methods, e.g. Substituted Cysteine Accessibility Method (SCAM) or biotinylation for probing protein conformational dynamics.

To clarify on the non-preliminary nature of the HDX-MS data, we have now inserted the following sentence in the Methods section line 548-549:

“The HDX-MS data presented in this paper is the average of at least three technical replicate measurements for each time point.”

Comment #2:

Pepsin is used for cleavage – could other proteases complement it?

Our response:

We thank the reviewer for the insightful suggestion. In fact, we did perform a screen of a wider range of HDX-MS compatible proteases, including recombinant rhizopuspepsin, nepenthesin I and nepenthesin II. We found that for this setup, the other proteases did not complement pepsin with additional sequence coverage.

Comment #3:

Furthermore, the study is performed in detergent, not in a membranous environment, so subtle details of dynamics and co-operative movements are very likely flawed by the detergent-solubilized state of the protein. Line 326 attempts a justification of detergent-to-lipid bilayer analogy by comparing it to LeuT studies in either lipid or detergent – this is not a valid comparison and validation of the current study. The results invite a much wider use of MD simulations (past literature and new studies/analyses) to reach further qualification and mechanistic insight from the models of dynamics proposed and how they affect the transport activity overall.

Our response:

We agree that our experimental setup only models and not mirrors the situation with DAT in its native environment. A membranous environment, such as a constructed lipid bilayer (e.g. nanodiscs or liposomes), would also constitute a model (although likely a better one) relative to DAT expressed in a neuron. We also agree that *‘subtle details of dynamics and co-operative movements are very likely flawed by the detergent-solubilized state of the protein’*, to the degree that the dynamics observed in detergent may be more pronounced for some domains relative to a lipid bilayer. This has been observed in our previous study on LeuT as well as on a recent study on the secondary transporters LacY, Xyle and GlpT (Martens et al. 2018, Nat. Commun.). Importantly, however, for all mentioned proteins, the overall HDX profile and impact of ligand binding was very similar between the detergent and lipid model systems. We agree that this observation from other transport proteins does not fully validate a similar comparison for dDAT. However, it does suggest that DDM micelles are a valid model for other membrane proteins with the same fold and functionality as dDAT. Also, our current controls suggest that our dDAT preparation follows what is usually ‘the golden standard’ for determination of protein integrity: (I) It binds [³H]Nisoxetine and dopamine with similar affinities as when situated in a membrane, suggesting a preservation of overall fold and stability. (II) We observe dynamic changes in domains, which are expected to be involved in the transport process. (III) A large amount of lipid and cholesteryl hemisuccinate is present in our samples,

which is essential for DAT functionality, suggesting that these molecules are organized around vital parts of the protein. However, we can never be 100% certain that dDAT behaves exactly the same way when embedded in the lipid bilayer of the neuron. Anything diverging from that setting would be an approximation.

To further accommodate this very relevant point, and meet the suggestion by the reviewer on using MD simulations, we have now performed extensive 500 ns all-atom MD simulations of dDAT wild-type embedded in a DDM micelle and in a mixed lipid bilayer. We find that the root-mean-square fluctuations (RMSF) for the simulations of dDAT in the DDM micelle and the lipid bilayer are very similar (see new Fig. 1a,b,c) indicating that the two systems display similar structural dynamics. This suggests that, although there are minor differences, the two system setups are overall comparable. For further details, please see the entire new section in Results describing the MD simulations, including a new Figure 1.

Again, we thank the reviewer for this valuable suggestion. Indeed, we think that this new section has considerably strengthened the entire foundation for performing the HDX-MS experiments.

Comment #4:

Line 56: "...first insights into the conformational dynamics underlying substrate binding and their possible role...". This is not a fair justification and qualification of the study given the enormous literature on NSS transport and dynamics

Our response:

We agree that it was a bold statement. We have changed it accordingly to specify. The sentence is changed to (line 60-62):

"Our results provide direct experimental insights into the conformational dynamics underlying substrate binding and isomerization between functional states of DAT"

Comment #5:

Generally the manuscript is difficult to follow, and needs a substantial rewriting and mechanistic focus.

Our response:

We thank the reviewer for the comment. We have now performed an overall revision to increase its readability and its mechanistic focus.

Reviewer #2 (Remarks to the Author):

In this article, the authors investigate the conformational dynamics of the human dopamine transporter DAT. They use HDX-MS in three different conditions (apo, with Na⁺, with Na⁺ and substrate dopamine) to identify the regions involved in the structural rearrangements underpinning the transport cycle. Their findings seem to overall confirm the rocking-bundle alternating-access mechanism despite some

discrepancies. They identify a region hugely stabilized upon ligand binding which was not previously pointed out as mechanistically relevant.

Overall, it is a well-written manuscript, with a solid experimental set-up, which gives novel insights into the mechanism of an important drug target and will lay the foundation for other structural studies.

Our response:

We thank the reviewer for complementing the manuscript as well written, having a solid experimental setup and providing novel insights into the mechanism of an important drug target. We too expect that our findings will lay the foundation for other structural studies.

Major concerns

Comment #1:

My main concern is whether the EX1 kinetics observed might be indicative of a local unfolding that is not physiologically relevant but it may be just caused by the protein degrading over time during the HDX experiment which is conducted at 25C. This should be tested and reported in the manuscript. The authors could consider conducting some parallel experiments (e.g. FTIR, CD or analytical SEC) to assess the stability of the transporter over time. In line 230 the authors suggest they actually have some data on stability but they don't elaborate further – they should show this data. Since eukaryotic transporters are notoriously unstable, such control is essential. Most of this EX1 kinetics were observed in the apo state, which I believe is the most unstable.

Our response:

We thank the reviewer for pointing this out and appreciate this opportunity to address this important point. As suggested, we have now also performed analytical SEC to assess changes to the monomeric state of DAT during the incubation time used during HDX-MS. Accordingly, dDAT was incubated in similar buffers as for the HDX experiments, either 200 mM Cs⁺ or 200 mM Na⁺ for either 0, 1, 4 or 8 hours, followed by SEC. Our data shows that dDAT is very stable in Na⁺ showing no apparent multimerization or aggregation over the 8 hour period. We do find that dDAT slowly multimerizes exclusively in the Cs⁺ (apo state) with approx. 30% decrease in the monomeric peak after 8 hours and concomitant increase in oligomeric/void volume fraction.

We are confident that we can conclude that the observed EX1 kinetics is not correlated with this slow multimerization of the apo state of DAT. First, the EX1 kinetics in all regions of the apo state (apart from TM12) are much faster than the rate of multimerization observed by SEC, i.e. we have approx. 100% of interconversion to the high-mass fraction significantly before 8 hours e.g. TM6b-IL3-TM7 fully converts within 10 min in the apo state and most other regions even faster (Figure 5 and Table 1). Second, for EL2 (peptide 215-222) and the C-helix (peptide 577-593), we also observe EX1 kinetics in the Na⁺-state (and the Na⁺/DA state for that matter). We did not find it relevant to perform analytical SEC for the Na⁺/DA sample since DA causes a further stabilization relative to Na⁺ alone.

The EX1 kinetic behavior in TM12 shows approx. 30% of the high-mass population after 8 hours, which could in principle correlate with an increased amount of multimerized dDAT. However, any precipitated dDAT would be removed before pepsin cleavage and MS analysis. Furthermore, in order for irreversible

multimerization to disguise itself as EX1 kinetics in TM12, it would then need to be due to local unfolding in a population of the protein over time. The remaining monomeric population would however likely exchange according to EX2. However, in TM12 we observe that the low mass population does not shift along the m/z-axis within the entire 8-hour period, meaning that exchange appears to occur exclusively via EX1 and not EX2.

In order to elaborate on this and modify our interpretation of the EX1 kinetic behavior in TM12 to include a disclaimer, we have now modified the results section.

We have inserted the following in the Results section (line 322-343):

“To ensure sample stability of dDAT during the 8 hour time-frame of the HDX-MS experiment, we performed analytical size-exclusion chromatography (SEC) of the apo state and the Na⁺-bound state after incubation at 25°C in similar buffers as for the HDX experiments and matching time intervals (Supplementary Fig. 7). Analytical SEC showed that dDAT in Na⁺ is very stable and remains monomeric showing no formation of multimers over the 8 hour period. We did observe that dDAT slowly formed multimers exclusively in the apo state with approx. 30% decrease in the monomeric peak after 8 hours and concomitant increase in the oligomeric/void volume fraction. Importantly, our results do not indicate that the EX1 kinetics observed in both the apo, Na⁺-bound, and Na⁺+DA-bound states are related to the multimerization observed for the apo state. First, the Na⁺-bound state of dDAT remains fully monomeric during the HDX-MS time-frame, yet we observe EX1 kinetics in both this and the Na⁺+DA-bound state. Second, the EX1 kinetics observed in all regions of dDAT (apart from TM12) are much faster than the rate of multimerization observed by SEC, i.e. we have approx. 100% of interconversion to the high-mass fraction significantly before 8 hours, e.g. TM6b-IL3-TM7 fully converts within 10 min in the apo state and most other regions even faster (Fig. 5, Table 1). Third, the EX1 kinetic behavior in TM12 shows approx. 30% of the high-mass population after 8 hours (Supplementary Fig. 6d,e), which could in principle correlate with the increased amount of multimerized dDAT. However, any precipitated dDAT resulting from multimerization would be removed during the filtration step before online pepsin cleavage and MS analysis. Furthermore, in order for irreversible multimerization to disguise itself as EX1 kinetics in TM12, it would then need to be due to a local irreversible unfolding of TM12 in a population of the protein over time. The remaining monomeric population would however likely exchange according to EX2. The low-mass population in TM12 does not detectably shift along the m/z axis within the entire 8-hour period, thus indicating that exchange observed in TM12 occurs exclusively via an EX1 mechanism.”

- and in the Discussion:

“Another possibility is that it could be a result of local irreversible destabilization, especially because the most pronounced EX1 is observed in the apo state, which is the most unstable state. Indeed, our SEC shows that monomeric dDAT in the apo state is slowly converted into multimers and as a result, the monomeric fraction falls to approx. 66% after 8 hours. However, dDAT in the Na⁺-bound state exhibits no multimerization or change in stability even after 8 hours, yet we still see pronounced EX1 kinetics in both the Na⁺-bound and Na⁺+DA-bound states. Furthermore, for all domains exhibiting EX1 apart from TM12, the exchange to the fully deuterated high-mass isotopic envelope happens much faster ($t_{1/2} < 15$ min) than the partial formation of multimers observed exclusively for the apo state. Overall, this suggests that there is

no correlation between the slow destabilization of the dDAT apo state and the EX1 kinetics observed across all DAT states.”

Comment #2:

I find the results section difficult to read and partially confusing. Moreover, the figures could be improved, e.g in figure 1, each highlighted peptide could be represented individually on a cartoon version of dDAT and added as an inset in the uptake plots. Please annotate the topological maps with helix numbers, and indicate where the intracellular and extracellular regions are.

Our response:

We thank the reviewer for the comment. The figures have existed in multiple versions and this suggestion improved them even further.

We have updated figure 2 (previously figure 1) to show the sequence coverage highlighted on the crystal structure (figure 2a) and, as suggested, added an inset to each of the uptake plots to show the location of the individual peptides. We have also annotated the topology maps in figure 3, 4, 5, and Supplementary figure 44 (previously figure 2, 3, 4 and Supplementary figure 4) and indicated the intracellular and extracellular regions in all figures. In our view, this has significantly improved the readability of the figures.

Comment #3:

Could the authors explain why they add lipids in the buffers? Is it common practice for dDAT purification? The presence of lipids is not trivial because it has been shown to modulate the conformational dynamics of other transporters and receptors, even in the absence of a bilayer (doi: 10.1038/nchembio.1960, doi: 10.1038/nsmb.3262). The authors might want to discuss this possibility with respect to their findings.

Our response:

The buffer composition is the same as used by Gouaux and coworkers for purification of the dDAT constructs used for crystallization. We have found that the presence of lipids and CHS is absolutely essential for dDAT functionality. We are positive that it will modulate the conformational dynamics of dDAT, but we expect it to be closer to its physiological state. Probably, lipids are coordinated at specific domains of dDAT that cannot be replaced by DDM, in much the same way as cholesterol has its unique binding sites. It is possible that the dDAT micelles do have a bilayer-like appearance.

As suggested by the reviewer, we have inserted the following in the Discussion (Line 399-403):

“Indeed, the presence of lipids and CHS were absolutely essential for dDAT functionality and they will most likely contribute to the conformational dynamics of dDAT, possibly bringing it closer to its physiological state in a lipid bilayer. Clearly, the added lipids are coordinated at parts of dDAT that cannot be replaced by DDM in much the same way as cholesterol has its unique binding sites¹⁰.”

Other comments

Comment #4:

The use of the word “novel” twice in the same sentence – one time must be enough.

Our response:

We agree. This is now corrected.

Comment #5:

Line 70: Could the authors elaborate a bit on the optimization of their purification protocol? Is the use of mixed micelles of detergent and lipids a common practice for the purification of mammalian transporters?

Our response:

As explained above, this is ‘standard’ practice (if one can, at this early stage, talk about a standard for purification of eukaryotic transporters). It has shown to work for both dDAT and hSERT. Excluding lipids and CHS leaves both transporters unable to bind ligands. We have elaborated on this by adding the following sentence in continuation of the sentence referenced above (line 112-115):

“The presence of the cholesterol analog CHS and lipids in combination with DDM in the buffer increases the resemblance of the membrane mimic to a lipid bilayer and proved essential to preserve dDAT functionality throughout the purification procedure in agreement with previous reports^{10,13}.”

Comment #6:

Fig.1: What is the meaning of the color gradient of the cyan peptides? This seems a bit confusing.

Our response:

We thank the reviewer for this comment. It was an attempt to show sequence coverage and changes in stability at the same time. We agree that this information may be lost and there are problems with distinguishing the colors. We have now updated the figure (now figure 2) to avoid confusion. The cyan/grey now distinguishes only between sequences with and without coverage, respectively. See also our reply to your comment #2.

Comment #7:

Line: 140 “Interestingly, the destabilizing effects in TM1b and TM7 contrast the stabilizing effects in TM6a and TM2.” Could the authors elaborate on this observation? Why is it interesting? Is it interesting because it doesn’t fit with what the authors expected from an outward-open conformation?

Our response:

We welcome the change to clarify. We have rephrased the sentence as follows to elaborate on why this observation is interesting (line 185-192):

"Interestingly, the destabilizing and stabilizing effects seen in the bundle domain do not correlate with our observations from LeuT²⁷, where Na⁺ stabilized the intracellular parts and destabilized the extracellular regions. Here, stabilization and destabilization do not show a clear pattern: the extracellular parts of TM1 and TM7 are destabilized, but stabilized in TM2 and TM6. A different pattern is seen on the intracellular side, where TM1 and TM6 are stabilized, TM7 destabilized and no effect in TM2. The asymmetric conformational dynamics indicate that the bundle domain does not necessarily operate as a rigid bundle as suggested for LeuT^{18,19}, but rather mediate the transitions in the TM-domains that facilitate substrate binding and subsequent translocation."

Comment #8:

Line 142: "Similar effects were seen for EL4 in LeuT when comparing the outward-favoring condition with an inward-favoring LeuT mutant, indicating that ion binding favors the outward-open conformation of dDAT, with TM7 and EL4 participating in the opening of the extracellular vestibule". I find this sentence confusing: how do LeuT results support the fact that dDAT is in an outward open conformation upon sodium binding?

Our response:

We missed a couple of full stops here. The sentence has been rephrased. It now reads (line 194-198):

"EL4 in LeuT exhibited a similar HDX profile between the outward-favoring condition and an inward-favoring mutant²³. Together, this suggests that, as in LeuT, Na⁺ binding favors the outward-open conformation of dDAT, with TM7 and EL4 participating in the opening of the extracellular vestibule."

Comment #9:

Figure 2: the panel b is very difficult to read. Maybe the authors could show only stabilized peptides on one structure and only destabilized on another structure? And colour-coded the peptide ID as well?

Fig.3. Same as above. Panel b is very hard to read and panel c should be annotated.

Our response:

We thank the reviewer for these comments. We agree that the readability of panel b in figure 2 and 3 (now figure 3 and 4) could be improved. Therefore, we have now simplified panel b by removing the arrows and the corresponding text indicating the residue numbers of the stabilized and destabilized regions on the structure. Instead, we have only indicated the structural motifs on the crystal structure. To further simplify panel b, we have removed the dotted lines that displayed the wild-type dDAT regions not resolved in the crystal structure/truncated in the crystallization construct. In addition, we have added an approximation of the membrane location in panel b. In panel c, we have annotated the topology as well as indicated the C-terminal tag with black non-filled circles in the snake diagram. We think that these changes have significantly improved the readability of the figures. We have made the same changes to Supplementary figure 4.

Comment #10:

Discussion: The discussion pretty much suggests that at least part of the results will be very different in a lipid bilayer. Have the authors carried out MD simulations to provide some hints on how much different things could be?

Our response:

We are not sure where this is explicitly suggested in the original version of the Discussion. However, based on several comments from the reviewers we have now specifically addressed this question by performing MD simulations. We find it entirely possible that the conformational dynamics are different for dDAT in its natural neuronal environment, with a specific lipid composition and associated proteins etc. The question is how good a model our DDM/lipid/CHS micelles are. Please also see our response to Reviewer #1's Comment #3 and Reviewer #3's Comment #1. Here, we also explain the outcome of our all-atom MD simulations. They support our micellar preparations as a valid model for dDAT dynamics in a lipid bilayer. Taken together, we believe that our model comes quite close to the physiologic effects of dDAT in a bilayer, although there would likely be subtle differences. Possibly, we would see similar changes in dynamics in the same protein domains, although of a lesser magnitude.

Comment #11:

Supplementary

- Extended data figure 2. Not sure I understand the purpose of panel b. The representative spectra are shown at only one time point. This doesn't match the legend.

Our response:

We thank the reviewer for pointing out the inconsistency between the figure and the figure legend. The purpose of panel b is to exemplify the right-shift of the isotopic envelope to higher m/z-value as function of time and ion/substrate composition. We have updated the legend to match the figure.

The explanation for panel **b** now reads (Supplementary Information line 32-34):

“**b**, Representative mass spectra for peptide 37-43 from dDAT for undeuterated (0 min) and after 10 min of deuterium exchange. The isotopic envelope shifts to different m/z values as a function of added Na⁺ and DA due to the respective degree of deuterium incorporation.”

- Thanks for providing the uptake plots, that's very useful.

Our response:

You are welcome 😊

Reviewer #3 (Remarks to the Author):

The manuscript by Nielsen et al. seeks to map the conformational dynamics of the dopamine transporter by substrate and sodium ions. This is an interesting manuscript and exciting study that provides insights into the functioning of the neurotransmitter: sodium symporters. However, there are several questions that I have that relate to the methodology and interpretation that needs to be better explained in a revision before the manuscript can be accepted for publication. These are highlighted below:

Our response:

We appreciate that the reviewer finds our manuscript interesting and exciting.

Major points:

Comment #1:

My concern is that the dopamine transporter in this study is detergent solubilized and even though is examined in a mixture of micelles, it is still not embedded in a lipid matrix. Given this, it is unclear how it can be concluded that “Our HDX-MS data provides the first spatially-resolved view of the conformational dynamics of DAT”. The authors should more clearly describe how anchoring in lipid might alter/enhance function of the transporter.

Our response:

We agree that it would be a better model of DAT conformational dynamics if they were performed in a lipid matrix. However, our work herein is still an experimental description of DAT conformational dynamics that is the closest model to a natural biological setting performed to date. We constantly stride for better models, but applying a lipid matrix (liposomes, nanodiscs etc.) imposes additional substantial challenges to the sample preparation and readout. We are not aware of any other laboratory that has accomplished this for DAT yet. We have ensured DAT functionality by showing that it binds ligands (dopamine and nisoxetine) with affinities similar to those reported for dDAT expressed in cells and that it is relatively stable during the HDX-MS experiments. Please also see our response to Reviewer #1's Comment #3.

To further accommodate this relevant comment, we have performed all-atom MD simulations of dDAT in detergent micelles and a lipid matrix. The modelling shows similar dDAT dynamics in the two setups, with the difference that the dynamic domains are slightly more flexible in detergent micelles. In all, this suggests that DDM/CHOL micelles is a good membrane mimic for dDAT. To include this, we have added a completely new section in results with MD simulations, a new Fig. 1 and Methods section. Please see Results section for this extensive addition.

The additional results have been further substantiated by recent publications from Politis and co-workers showing overall similar HDX pattern for the Xylose transporter in DDM and nanodiscs (Martens et al. 2018, Nat Commun) as well as in our publication comparing HDX of LeuT embedded in either DDM or nanodiscs (Merkle et al 2018, Sci. Adv.).

Comment #2:

Dopamine transport is directional. It is unclear how a mixture of inside-out and outside-in orientation of transporters in detergent micelles, can allow correlation of dynamics as read out by HDX with function?

Our response:

The micelles organize and stabilize the hydrophobic areas of dDAT, which normally are supported by lipids i.e. mainly the transmembrane domains. The detergent is not expected to cover or obscure the intracellular or extracellular domains. They will not form droplets as liposomes will. Hence, detergent will allow access to both sides of the transporter at once, much as in a nanodisc. What we observe is a change in the conformational dynamics upon the binding of ligands to distinct binding sites in the transporter (Na⁺ and DA) – and as such the effects are independent of the orientation of DAT in the detergent micelle. We agree that our experiments do not fully mimic the in vivo lipid environment of DAT in neurons, please see also our replies to Comments #3 and #10 by Reviewers #1 and #2, respectively.

Comment #3:

In figure 1, the Y-axis should be average uptake of deuterons. Not clear what is meant by Relative Uptake? Relative to what? It seems like absolute exchange.

Our response:

We agree that 'Deuterium uptake' is a more accurate term. We have now changed the Y-axis to 'Deuterium Uptake' in figure 1 (now figure 2b) as well as in Supplementary Figure 3 and throughout the text in the manuscript as it appears from the track changes.

Comment #4:

According to the authors as stated in the abstract, "slow cooperative fluctuations are observable in several domains of the transporter, which could be a novel molecular mechanism that assists in the transporter function" I am not clear how these slow fluctuations contribute to transporter function.

Our response:

We agree that this is a rather implicit statement. It is an attempt to be as scientifically objective as possible without any mechanistic interpretation. 'Slow cooperative fluctuations' refer to EX1 kinetics, which is observed when a region of the protein undergoes exchange by a concerted breakage of a stretch of backbone hydrogen bonds resulting in an open and exchange-competent conformation, which refolds or closes at a rate (k_{cl}) that is significantly slower than the chemical exchange rate (k_{ch}). We would like to refrain from writing this detailed interpretation in the abstract. Instead we have now attempted to be more explicit in the Introduction section, by rephrasing the sentence here to (line 64-69):

"In addition, we find that several helical regions of dDAT undergo slow cooperative fluctuations that break multiple backbone hydrogen bonds resulting in an open and exchange-competent conformation, which refolds or closes at a rate that is significantly slower than the chemical exchange rate. These slow concerted

fluctuations of helices are modulated by ligand binding, suggesting them to be an intrinsic part of the transport mechanism.”

We elaborated even further in the Results section (see also our reply to your comment #8). To further meet this valid point, we also performed a general revision of the wording in the Discussion section. It now reads (line 413-432):

“Slow cooperative fluctuations, or EX1 kinetics, are rare observations in proteins, but have also been observed in transporters such as P-glycoprotein⁴⁸, G-protein coupled receptors⁴⁹ and in enzymes⁶¹. EX1 behavior arises from correlated protein motions that occur slowly relative to the rate of the chemical exchange reaction. Our observation of EX1 in several TM helical regions of DAT is in good correlation with our earlier observations from LeuT²⁷ and can be interpreted as the hydrogen-bonds of the helical segment breaking in a slow concerted manner, i.e. a partial unwinding or bending of the helical structure. This partially unwound helix stays open or unwound in a sufficient time-frame to allow several amides to exchange their hydrogens in a correlated manner (i.e. $k_{cl} \gg k_{ch}$) before closing again. Interestingly, the observation that the EX1 kinetics was attenuated by ligand binding could suggest that partial helix unwinding in the observed domains is an integrated part of the translocation mechanism. Another possibility is that it could be a result of local irreversible destabilization, especially because the most pronounced EX1 is observed in the apo state, which is the most unstable state. Indeed, our SEC shows that monomeric dDAT in the apo state is slowly converted into multimers and as a result, the monomeric fraction falls to approx. 66% after 8 hours. However, dDAT in the Na⁺-bound state exhibits no multimerization or change in stability even after 8 hours, yet we still see pronounced EX1 kinetics in both the Na⁺-bound and Na⁺+DA-bound states. Furthermore, for all domains exhibiting EX1 apart from TM12, the exchange to the fully deuterated high-mass isotopic envelope happens much faster ($t_{1/2} < 15$ min) than the partial formation of multimers observed exclusively for the apo state. Overall, this suggests that there is no correlation between the slow destabilization of the dDAT apo state and the EX1 kinetics observed across all DAT states.”

Please also see our response to Comment #8 and our revised explanation of EX1 kinetics in the Results section on Cooperative fluctuations in dDAT helical domains (see track changes in the document).

Comment #5

The Km of dopamine is 8.2 uM according to reference 13 (Wang et al. (2013)), but the Ki here reported is 1.7 uM. The authors need to explain what nisoxetine is. The papers cited are measuring dopamine transport and so are reporting Km values but here the authors are reporting displacement. The authors need to clarify.

Our response:

We thank the reviewer for this comment. First, we have now adjusted line 118 to explain what nisoxetine is.

“Purified dDAT bound the high-affinity inhibitor [³H]nisoxetine with a dissociation constant (K_d) of 80 ± 6 nM in the presence of 200 mM NaCl.”

Second, the wild-type dDAT (dDAT_{WT}) exhibits DA transport with a K_M of 2.1 μ M according to reference 13 (Wang et al. 2015, Nature), while the thermostabilized mutant, which Wang et al. refer to as the minimal functional construct (dDAT_{mfc}), transports DA with a K_M of 8.2 μ M. We report an affinity (K_i) of DA to dDAT_{WT} of 1.7 μ M, which was determined by displacement of the inhibitor [³H]nisoxetine by DA. Both K_M and K_i represent affinities, and we therefore compare the affinity of DA to dDAT measured by DA transport in whole cells to the affinity of DA to dDAT measured by competitive inhibition of [³H]nisoxetine to purified dDAT.

Comment #6:

On page 6, line 104, it is not clear what is meant by “the majority of identified dDAT peptides showed a gradual increase in deuterium uptake as a function of time, indicating the presence of secondary structured regions in good agreement with the structural folds observed in the crystal structures” How is a gradual increase in deuterium uptake as a function of time correlated with secondary structured regions? How is this being inferred?

Our response:

Backbone amide hydrogens present in disordered regions of a protein, where they do not participate in hydrogen bonding, are expected to exchange very quickly for deuterium due to the unrestricted access to the exchange reaction site. However, in secondary structured regions of a protein, the backbone amide hydrogens participate in hydrogen bonding and are therefore not in an exchange-competent state. However, naturally occurring fluctuations in the secondary structure, where the hydrogen bonds transiently break, will lead to exchange of the backbone amide hydrogens for deuterium, and hence, a gradual increase in the deuterium uptake. As a result, peptides covering secondary structure in the intact protein will exchange its backbone amide hydrogens much slower than observed for unstructured regions. We have modified the sentence as follows to clarify this (line 149-154):

“The majority of identified dDAT peptides showed a gradual increase in deuterium uptake as a function of time indicating the presence of secondary structured regions, while the N- and C-termini as well as major parts of EL2 displayed maximum uptake after the first sampled time point (15 sec) suggesting unstructured regions, which is in good agreement with the structural fold observed in the crystal structures^{10,13,14} (Fig. 2b; Supplementary Fig. 3).”

Comment #7:

In figure 1, the red chlorine atom and the second sodium ion are not evident. The cyan gradation is not explained. TM1b for instance is a darker shade of cyan. How much a difference separates the dark cyan from light cyan? It is also not useful to highlight peptides showing difference in uptake between at least two states. What time points are being considered. Is it the maximal difference at any timepoint that it may be at? For instance, for peptide 320-328 (TM6 hinge), are you representing a difference of 3.0 D (for apo)

minus 0.5 D (for Na⁺ + DA) at 5000 min as your dark cyan. How is this dark cyan equivalent to peptide 48-56 (TM1b) where Na⁺ + DA shows greater exchange than apo. These two peptides show opposing effects of sodium and DA and so the cyan coloration of the structure is misleading and uninformative!

Our response:

We thank the reviewer for the very useful comment. We have completely revised Figure 1 (now figure 2). The Cl⁻ ion and the two Na⁺ ions are now evident. In addition, we have rewritten the figure legend. The cyan gradation now only depicts areas of dDAT where we have sequence coverage. We have highlighted the peptides on an inset of the topology of dDAT in each uptake plot, which hopefully is more clear and informative.

The legend now reads (line 806-818):

“Figure 2. HDX of local regions of dDAT in different functional states. **a**, Cylindrical representation of the crystal structure of dDAT in complex with two Na⁺ ions (blue spheres), one Cl⁻ ion (red sphere) and DA (orange spheres) (PDB ID: 4XP1). To obtain information about HDX of local regions of dDAT, we digested the transporter enzymatically and identified the generated peptides using MS. Regions of dDAT covered by identified peptides are highlighted in cyan. Wild-type dDAT was used for the HDX-MS experiments, however, the crystal structure shown has truncated N- and C-termini and deleted part of EL2. The C-terminal His-tag used for purification of the wild-type dDAT is added to the structure as circles with one-letter codes. **b**, Deuterium uptake is plotted as a function of labeling time (*i.e.* 0.25 min – 480 min) for representative peptides of dDAT. Red, blue and orange curves illustrate the deuterium uptake for dDAT in the apo, Na⁺-bound, and Na⁺/DA-bound state, respectively. Values represent means of three (for apo and Na⁺/DA) or six (Na⁺) independent measurements. Standard deviations are plotted as error bars but are in most instances too small to be visible. Maximum-labeled control samples are shown as black circles at 1440 min. Insets: The location of the corresponding peptide is highlighted in black on the topology map of dDAT.”

Comment #8:

The entire results section on the Cooperative fluctuations in dDAT helical domains needs to be revised. It is more of an observation. It is not clear how sodium contributes to these cooperative fluctuations and how these might contribute to transporter function.

Our response:

We appreciate this comment. The section has been challenging to write in a clear language without too much interpretation. We have performed a substantial and general revision of the section. This revision also includes Reviewer #2's comment #1. In particular, we have replaced the introduction to the section to provide a more explicit explanation of the biological differences between EX2 and EX1 kinetic reactions. We hope that it provides a clearer picture of the biological relevance. In addition, we have added sentences to put perspectives on the observations. Importantly, we have also rewritten most of the discussion to increase the readability (see also above Comment #4). Apart from the general revision (see track changes in document), we have added the following (line 241-258):

“In an HDX-MS experiment, the exchange between hydrogens and deuteriums most commonly occurs through an EX2 time regime in which the exchanging protein segment undergoes opening and closing dynamics that occur at rates (k_{op} and k_{cl}) that are significantly faster than the rate of the chemical exchange reaction k_{ch} (specifically, $k_{cl} \gg k_{ch}$)^{40,47}. Thus, the exchange of individual backbone amide hydrogens in this segment are uncorrelated the observed rate of HDX (k_{HDX}) reports on the stability of local hydrogen-bonded structure in this segment. Such uncorrelated exchange appears as a gradually increasing binomially distributed isotopic envelope in the mass spectrum (see Supplementary Fig. 5a for example). While the majority of proteins exchange with EX2 kinetics under physiological conditions, there are increasing numbers of membrane proteins with reported EX1 kinetics^{27,48,49}. In contrast to EX2 kinetics, EX1 kinetics is characterized by concerted opening and closing motions of several backbone amide hydrogen bonds that are slower than the rate of the chemical exchange (specifically, $k_{cl} \ll k_{ch}$). In the mass spectra, this is seen as a bimodal isotopic pattern resulting from the presence of a low-mass population, which has not yet undergone the cooperative fluctuating motion required for exchange, and a high-mass population in which the concerted opening event has occurred and multiple hydrogens have simultaneously exchanged for deuterium (Fig. 5; Supplementary Fig. 5b,6). In the EX1 time regime, the observed rate of HDX is equal to the rate of the concerted opening ($k_{HDX} = k_{op}$)^{40,47,50}. Thus, the presence of EX1 kinetics in a segment of a protein (e.g. a TM helix) allows a direct measure of the rate of conformational opening and the half-life of the closed/folded state ($t_{1/2}$).”

Please also see our response to Comment #4 and our addition to the Discussion section for further explanation on EX1 kinetics.

Comment #9:

In Extended figure 1, why was dDAT deglycosylated? When it is stated that the glycosylation was retained on the protein for effective function (line 73).

Our response:

We appreciate the opportunity from the reviewer to clarify this. As stated, the glycosylation was retained on dDAT to avoid possible deviation from its native conformational properties. However, we show deglycosylated dDAT on the SDS-PAGE, lane 3 in Extended Data Figure 1 to illustrate that the two bands seen in lane 2 correspond to heterogeneously glycosylated dDAT, and hence, we show lane 3 to avoid any misunderstandings with regards to the purity of the sample. We have now updated the figure text in order for this to be clear to the reader:

“**Supplementary Figure 1. Purification and functional characterization of dDAT. a,** Representative SDS-PAGE showing purity of dDAT eluted from nickel immobilized-metal affinity chromatography. Lane 1, solubilized material before incubation with Ni-NTA resin. Lane 2 and 3 shows the purity of dDAT eluted from Ni-NTA resin. The two major bands in lane 2 are caused by heterogeneous glycosylation of dDAT as evident from lane 3, where dDAT appears as one single band after the same sample has been deglycosylated with PNGaseF. However, for the following binding and HDX-MS experiments, dDAT was left glycosylated to retain as wild-type like functionality as possible.”

Comment #10:

What is meant by 'POTENTIAL EX1/EXX kinetics' in Figure 4?

Our response:

EX1/EXX kinetics means that for these peptides we observe a broadening of the isotopic envelope which is not readily resolved into two distinct low-mass and high-mass isotopic envelopes. Accordingly, it could be a mixture of EX1 and EX2 kinetics, known as EXX or EX1 kinetics involving only a few amide hydrogens. We did provide an explanation for this in the Results section, but failed to do so in the legend for Figure 4 (now 5). The first part of Fig. 5 legends now reads:

“Figure 5. Correlated exchange kinetics of regions in dDAT. a,b Regions in dDAT, for which EX1 or EXX (a mixture of EX1 and EX2) kinetics were observed, are marked in dark green on the crystal structure (a) (PDB ID: 4XP1) and snake diagram (b) of dDAT.”

'Potential' refers to the fact that we observed signs of EX1/EXX kinetics, but as stated in the Results section “a reliable quantitative bimodal deconvolution was not possible for peptides spanning these regions”. We have updated figure 4 (now figure 5) with “Potential EX1/EXX kinetics (uncharacterized)” and added “(referred to as 'potential EX1/EXX kinetics (uncharacterized) in Fig. 5)” in line 281-282 in the Results section to avoid any confusion.

Comment #11:

The parameters for ion mobility separations (line 400) or a reference should be provided.

Our response:

We thank the reviewer for pointing out the missing parameters for ion mobility separation. We have added the following to the Methods section (line 562-563):

“The ion mobility cell was operated using a constant nitrogen flow of 90 ml/min at a wave velocity of 580 m/s and a wave height of 40 V.”

Comment #12:

How was the mix of DDM + cholesteryl hemisuccinate (CHS) and lipids (POPC, POPE and POPG) derived?

Our response:

The buffer composition is the same as used for dDAT crystallization by Gouaux and coworkers (see e.g. Penmatsa et al. 2013, Nature). One could say it is the standard buffer (if one can use 'standard' at this early stage of purification of eukaryotic transporters). It has shown to work for both dDAT and hSERT. We have found that the presence of lipids and CHS is absolutely essential for dDAT functionality. Please also see our response to Reviewer #2's comment #3 and #5 addressing the same question.

Reviewers' Comments:

Reviewer #1:

Remarks to the Author:

The authors have done a great job of improving the analysis and the manuscript and responding to reviewer comments. I have no further comments

Reviewer #2:

Remarks to the Author:

The authors have carried out a thorough review leading to a much strengthen manuscript. In particular, I note the inclusion of additional MD simulations to compare the effect of detergents and a lipid bilayer on the conformational dynamics of DAT transporter as well as the significant improvements in the figures and the accompanied information pertaining the interpetation of HDX experiments. I therefore recommend the paper for publication.

Reviewer #3:

Remarks to the Author:

I am reasonably satisfied with the revisions made by the authors in response to the comments from all reviewers including my own. My biggest concern along with another reviewer was that the impact of the study was weaker when examining DAT in the absence of lipids. The authors have acknowledged this in their revision.